# IgM and IgD B cell receptors differentially respond to endogenous antigens and control B cell fate

Mark Noviski[1], James L Mueller[2], Anne Satterthwaite[3], Lee Ann Garrett-Sinha[4], Frank Brombacher[5,6], Julie Zikherman[2]*

[1]Biomedical Sciences (BMS) Graduate Program, University of California San Francisco, San Francisco, United States; [2]Rosalind Russell and Ephraim P. Engleman Arthritis Research Center, Division of Rheumatology, Department of Medicine, University of California San Francisco, San Francisco, United States; [3]Department of Immunology, UT Southwestern Medical Center, Dallas, United States; [4]Department of Biochemistry, University at Buffalo, The State University of New York, Buffalo, United States; [5]International Center for Genetic Engineering and Biotechnology (ICGEB), Cape Town, South Africa; [6]Institute of Infectious Diseases and Molecular Medicine, Division of Immunology, Faculty of Health Sciences, University of Cape Town & Medical Research Council (SAMRC), Cape Town, South Africa

*For correspondence:
julie.zikherman@ucsf.edu

Competing interests: The authors declare that no competing interests exist.

**Abstract** Naive B cells co-express two BCR isotypes, IgM and IgD, with identical antigen-binding domains but distinct constant regions. IgM but not IgD is downregulated on autoreactive B cells. Because these isotypes are presumed to be redundant, it is unknown how this could impose tolerance. We introduced the Nur77-eGFP reporter of BCR signaling into mice that express each BCR isotype alone. Despite signaling strongly in vitro, IgD is less sensitive than IgM to endogenous antigen in vivo and developmental fate decisions are skewed accordingly. IgD-only $Lyn^{-/-}$ B cells cannot generate autoantibodies and short-lived plasma cells (SLPCs) in vivo, a fate thought to be driven by intense BCR signaling induced by endogenous antigens. Similarly, IgD-only B cells generate normal germinal center, but impaired IgG1$^+$ SLPC responses to T-dependent immunization. We propose a role for IgD in maintaining the quiescence of autoreactive B cells and restricting their differentiation into autoantibody secreting cells.
DOI: https://doi.org/10.7554/eLife.35074.001

## Introduction

The pre-immune mature naïve B cell compartment must balance the need for a diverse antibody repertoire with the risk of autoantibody-mediated disease. Early in development, B cells randomly rearrange their immunoglobulin genes through VDJ recombination and encounter a series of tolerance checkpoints that serve to remove autoreactive B cell receptors (BCRs) from the repertoire. Strongly autoreactive immature B cells rearrange additional light chains to 'edit' their autoreactivity, and they ultimately undergo deletion if this is unsuccessful (*Shlomchik, 2008*). Yet, despite stringent counter-selection of autoreactivity, the mature follicular (Fo) B cell compartment retains cells reactive towards endogenous antigens (*Wardemann et al., 2003*; *Zikherman et al., 2012*). How these cells are restrained from mounting autoimmune responses is not fully understood.

We previously described a BAC Tg reporter mouse (Nur77-eGFP) in which GFP expression is under the control of the regulatory region of *Nr4a1*, an immediate early gene rapidly induced by antigen receptor signaling (*Mittelstadt and DeFranco, 1993*; *Winoto and Littman, 2002*). We showed that Nur77-eGFP expression in naïve B cells is proportional to strength of antigenic

**eLife digest** To defend an organism against invaders such as viruses and bacteria, cells of the immune system need to recognize and respond to foreign microbes. However, these immune cells must also avoid attacking 'self' – for example, the healthy tissues of the body – as this could lead to autoimmune disease.

B cells are a type of immune cell that is essential in order to produce antibodies, protective proteins that can identify and defend against a broad range of germs: in addition, certain antibodies can also recognize 'self'. When a B cell first develops, it places its antibody on its surface and uses this protein as a receptor (termed 'B cell receptor') to sense its surroundings.

Prior to mounting an immune response, B cells carry two closely related versions of the B cell receptor on their surface: IgM and IgD. Both IgM and IgD perform many of the same roles and can largely substitute for one another. However, B cells that recognize 'self' decrease their levels of IgM but keep high amounts of IgD on their surface. It is unclear why this is the case, but one possibility is that IgM and IgD may see 'self' differently.

To investigate this, Noviski et al. used mice with B cells that only carry either IgM or IgD, and tracked how these cells reacted to molecules from 'self' and foreign origins. IgM-only B cells reacted more strongly to 'self' molecules than IgD-only cells, which suggests that IgM is more sensitive to 'self' than IgD. In fact, in mice which are at risk for an autoimmune disease similar to lupus, deleting the IgM receptor prevented antibodies against 'self' from being produced. Therefore, reducing the amount of IgM receptors may be a way to keep B cells that are reactive against 'self' from inappropriately attacking the host. Meanwhile, the IgD receptor still allows B cells to mount protective antibody responses against foreign microbes.

Future studies are necessary to determine whether these differences between IgM and IgD also exist in humans. If this is the case, blocking the IgM receptor could become a new kind of treatment for certain autoimmune diseases.

DOI: https://doi.org/10.7554/eLife.35074.002

stimulation and consequently to autoreactivity (*Zikherman et al., 2012*). The most prominent characteristic of GFP$^{hi}$ reporter B cells is decreased surface IgM BCR relative to GFP$^{lo}$ cells. Goodnow and colleagues first suggested that IgM downregulation may mark autoreactive B cells in the normal mature repertoire shortly after they reported selective downregulation of IgM in the IgHEL (hen egg lysozyme) BCR Tg/soluble HEL Tg model system of B cell autoreactivity (*Goodnow et al., 1988*; *Goodnow et al., 1989*). Indeed, multiple studies in mice and humans have identified naturally occurring IgD$^+$IgM$^{lo}$ cells as autoreactive and 'anergic' or functionally unresponsive (*Duty et al., 2009*; *Kirchenbaum et al., 2014*; *Quách et al., 2011*; *Zikherman et al., 2012*). These results corroborate observations with several BCR transgenic model systems (*Cambier et al., 2007*). However, whether or how IgM downregulation might constrain autoreactivity remains unclear because naturally-occurring autoreactive B cells maintain high expression of the IgD BCR isotype (*Zikherman et al., 2012*).

IgM and IgD are splice isoforms of a common precursor heavy chain mRNA (*Moore et al., 1981*). While they differ in their Fc domains, both BCR isotypes contain the same antigen-binding domain, as well as identical 3-amino acid cytoplasmic tails, and they pair with Igα/β in order to initiate the canonical BCR signaling cascade (*Blum et al., 1993*; *Radaev et al., 2010*). However, the expression pattern of the isotypes differs; IgM expression begins as soon as heavy and light chains recombine early in B cell development, and persists until class switch recombination occurs following B cell activation (*Chen and Cerutti, 2010*). IgD is uniquely co-expressed with IgM during a narrow developmental window on late transitional and mature naïve Fo B cells as a result of alternate splicing regulated by the zinc-finger protein ZFP318 (*Enders et al., 2014*; *Pioli et al., 2014*). This suggests that IgD may play a critical role specifically in mature naïve B cells. Initial characterization of IgM- and IgD-deficient mice revealed only mild phenotypes and substantial redundancy; each isotype could mediate B cell development, initiate antibody responses to T-dependent and -independent immunization, and induce normal levels of steady-state serum IgG (*Lutz et al., 1998*; *Nitschke et al., 1993*; *Roes and Rajewsky, 1993*). This is consistent with prior studies in BCR

transgenic model systems demonstrating that each isotype alone can mediate B cell development, deletion, and activation (*Brink et al., 1992*).

IgM and IgD differ structurally; IgD has a much longer, flexible hinge region linking its Fab and Fc regions than IgM does (*Chen and Cerutti, 2010*). Recently, Ubelhart, Jumaa, and colleagues demonstrated that a short and inflexible hinge confers upon IgM the unique capacity to signal in response to monovalent antigens, while both isotypes can respond to multivalent antigens (*Übelhart et al., 2015*). However, the Goodnow lab subsequently showed that IgHEL Tg splenic B cells expressing either the IgD or IgM BCR exclusively could mobilize calcium in response to soluble HEL antigen in vitro, and each isotype can mediate a common gene expression program characteristic of anergy in vivo (*Sabouri et al., 2016*). Therefore, it remains unclear whether IgM and IgD BCRs expressed in an unrestricted B cell repertoire differentially sense bona fide endogenous antigens in vivo, particularly as the identity and nature of such antigens is largely unknown and not restricted to soluble monovalent antigens. For example, 5–10% of circulating naïve B cells in healthy humans express the well-characterized unmutated and autoreactive human heavy chain V-segment IGHV4-34, and these cells exhibit anergy, selective IgM downregulation, and recognize cell-surface antigens on erythrocytes and B cells in vivo (*Quách et al., 2011*; *Reed et al., 2016*).

We and others previously hypothesized that downregulation of IgM on Fo B cells might serve to restrain their response to endogenous antigen. Conversely, high expression of IgD on these cells might play a 'tolerogenic' role (*Zikherman et al., 2012*). To determine how IgM and IgD differentially regulate B cell responses to endogenous antigens, we generated Nur77-eGFP reporter mice deficient for either IgM or IgD (*Lutz et al., 1998*; *Nitschke et al., 1993*). Using this reporter of antigen-dependent signaling, we show that IgD is less sensitive than IgM to bona fide endogenous antigens in vivo despite higher surface expression and robust responsiveness to receptor ligation in vitro. Indeed, marginal zone (MZ) and B1a cells that express IgD but lack IgM induce less Nur77-eGFP than WT, and this is not attributable to repertoire differences. To further support these observations, we examine a series of cell fate decisions for which in vivo BCR signaling requirements have been previously defined. We show that IgD drives a pattern of development consistent with reduced endogenous antigen recognition, favoring MZ B cell fate and disfavoring B1a cell fate. Similarly, IgD alone is less efficient than IgM at driving SLPC expansion in response to endogenous antigens, a process that requires robust BCR signaling. However, IgD is sufficient for germinal center (GC) B cell differentiation. Our data suggest that reduced endogenous antigen sensing (i.e. signal transduction in response to antigen binding) by the IgD BCR isotype shunts autoreactive IgD$^{hi}$ IgM$^{lo}$ Fo B cells away from differentiation into SLPCs. We propose that predominant IgD expression maintains the quiescence of autoreactive B cells in response to chronic endogenous antigen stimulation, and limits autoantibody secretion in the context of rapid immune responses.

## Results

### Endogenous antigen is both necessary and sufficient for expression of Nur77-eGFP reporter in B cells in vivo

We previously showed that antigen recognition was both necessary and sufficient for Nur77-eGFP reporter expression by B cells in vivo (*Figure 1A*)(*Zikherman et al., 2012*). Thus, reporter expression reflects endogenous antigen recognition in vivo. Conversely, under steady-state conditions in vivo, the Nur77-eGFP reporter does not reflect signaling through other receptors expressed in B cells; indeed, loss of either CD40, or TLR3, 7, and 9 signaling has no effect on reporter expression in B cells in vivo (*Figure 1—figure supplement 1A*). Moreover, neither BAFF, nor IL-4, nor CXCR4-dependent signaling can regulate reporter expression in vitro (*Figure 1—figure supplement 1B–C*) (*Zikherman et al., 2012*). To further confirm that Nur77 expression in naïve B cells under steady state conditions is not regulated by microbial stimulation of pattern recognition receptors (e.g. TLRs 1, 2, 4, 6, and the TLR-4-like molecule RP105), we studied mice raised under either germ-free conditions or conventional specific-pathogen-free conditions. We observed no induction of endogenous Nur77 protein in splenic B cells or endogenous *Nr4a1* transcript in splenocytes in the presence of commensal flora (*Figure 1—figure supplement 1D,E*). Moreover, MyD88-deficient and MyD88-sufficient splenocytes and peritoneal B1a cells express comparable amounts of endogenous *Nr4a1* transcript and protein respectively under steady state conditions (*Figure 1—figure supplement 1F,G*).

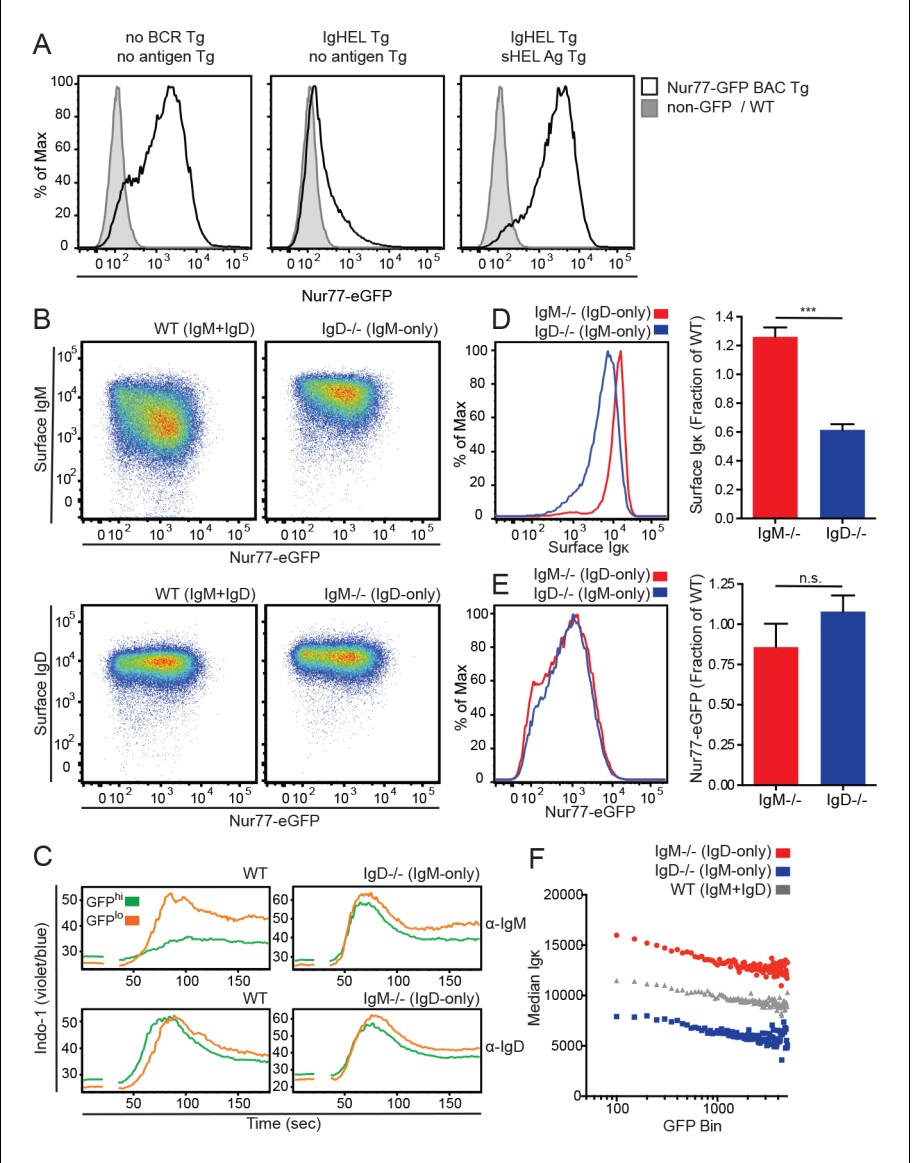

**Figure 1.** IgD expression enables a dynamic range of IgM responsiveness. (**A**) GFP expression in mature Fo splenic B cells (CD19+CD23+CD93-) from Nur77-eGFP BAC Tg reporter mice with either a wild-type BCR repertoire (left), or harboring IgHEL Tg specific for the cognate antigen HEL (hen egg lysozyme) in the absence (middle), or presence (right) of endogenous cognate antigen driven by soluble HEL Tg. WT Fo B cells lacking GFP reporter are included for reference (gray shaded histograms). (**B**) Surface IgM and IgD expression in splenic Fo B cells from WT, $IgM^{-/-}$, and $IgD^{-/-}$ mice expressing the Nur77-eGFP reporter. (**C**) Splenocytes from WT, $IgM^{-/-}$, and $IgD^{-/-}$ mice were loaded with Indo-1 and stimulated with 2.5 µg/mL of F(ab')$_2$ anti-IgM or 1:400 anti-IgD. Fo B cells with the highest 20% and lowest 20% Nur77-eGFP expression are compared. (**D**) Representative histograms and quantification of surface Igκ expression in $IgM^{-/-}$ and $IgD^{-/-}$ Fo B cells normalized to WT. (**E**) Representative histograms and quantification of Nur77-eGFP expression in $IgM^{-/-}$ and $IgD^{-/-}$ Fo B cells normalized to WT. (**F**) Median surface Igκ expression of WT, $IgM^{-/-}$, and $IgD^{-/-}$ Fo B cells was calculated for 200 bins of equal width across the Nur77-eGFP spectrum. For (A), (B) and (F), data are representative of at least n = 4 independent experiments. For (C), n = 3 independent experiments for anti-IgM and n = 2 independent experiments for anti-IgD. For (D) and (E), n = 7 and n = 4, respectively, WT, $IgM^{-/-}$, and $IgD^{-/-}$ mice. Welch's t test was used to calculate p values, and mean +SEM is displayed. ***p<0.001.

DOI: https://doi.org/10.7554/eLife.35074.003

The following source data and figure supplements are available for figure 1:

**Source data 1.** Numerical data corresponding to receptor and Nur77 reporter, protein and transcript levels in *Figure 1D–F* and *Figure 1—figure supplement 1D-G*, *2B-E*.

*Figure 1 continued on next page*

*Figure 1 continued*

DOI: https://doi.org/10.7554/eLife.35074.007

**Source data 2.** Numerical data corresponding to receptor levels in *Figure 1D*.

DOI: https://doi.org/10.7554/eLife.35074.008

**Source data 3.** Numerical data corresponding to Nur77-eGFP levels in *Figure 1E*.

DOI: https://doi.org/10.7554/eLife.35074.009

**Figure supplement 1.** Regulation of endogenous Nur77 and Nur77-eGFP reporter expression.

DOI: https://doi.org/10.7554/eLife.35074.004

**Figure supplement 2.** Quantification of surface BCR expression on B cell subsets.

DOI: https://doi.org/10.7554/eLife.35074.005

**Figure supplement 3.** Induction of Nur77-eGFP and CD69 in TLR-stimulated *IgM*$^{-/-}$ and *IgD*$^{-/-}$ B cells.

DOI: https://doi.org/10.7554/eLife.35074.006

Taken together, these data demonstrate that Nur77-eGFP expression in B cells under steady-state conditions in vivo is a *specific* readout of antigen-dependent signaling through the BCR (*Table 1*). We therefore sought to take advantage of the Nur77-eGFP reporter in order to probe the responsiveness of a diverse BCR repertoire of mature B cells expressing either IgM or IgD alone to the vast range of endogenous antigens they may encounter in vivo.

**Table 1.** Nur77-eGFP is a specific reporter of antigen-dependent signaling in vivo.

| Conclusion | Stimulus/ perturbation | Pathway | Readout | Cell type | References |
|---|---|---|---|---|---|
| Does not modulate Nur77 at steady state in vivo | *CD40L*$^{-/-}$ | CD40 | Nur77-eGFP | B cells | Manuscript *Figure 1—figure supplement 1A* |
| | *TLR7*$^{-/-}$ | TLR7 | Nur77-eGFP | B cells | Manuscript *Figure 1—figure supplement 1A* |
| | *Un93b1*$^{3d/3d}$ | TLR3/7/9 | Nur77-eGFP | B cells | Manuscript *Figure 1—figure supplement 1A* |
| | *MyD88*$^{fl/fl}$ MB1-Cre | MyD88 | Endog. Nur77 protein/transcript | PerC B1a cells/spleen | Manuscript *Figure 1—figure supplement 1F,G* |
| | Germ-free mice | MyD88/ TRIF | Endog. Nur77 protein/transcript | Splenic B cells/spleen | Manuscript *Figure 1—figure supplement 1D,E* |
| | Const. act. STAT5 | Jak/Stat | Nur77-eGFP | Thymocytes | Moran et al. JEM 2011, Figure 8. |
| Does not induce Nur77 in vitro | BAFF | BAFFR | Nur77-eGFP | B cells | *Zikherman et al. (2012)*: Figure S1G |
| | IL-4 | Jak/Stat | Nur77-eGFP | B cells | Manuscript *Figure 1—figure supplement 1B* |
| | IL-2, IL-15 | Jak/Stat | Nur77-eGFP | CD8 (IL-2, 15), CD4 (IL-2) | Au-Yeung et al. JI 2017, Figures S1A, 3C, 4A |
| | CXCL12/ SDF-1 | CXCR4 | Nur77-eGFP | B cells | Manuscript *Figure 1—figure supplement 1C* |
| Induces Nur77 in vitro but does not require IgM or IgD specifically | LPS | TLR4, Rp150 | Nur77-eGFP | B cells | *Zikherman et al. (2012)*: Figure S1G; Manuscript *Figure 1—figure supplement 3A* |
| | CpG | TLR9 | Nur77-eGFP | B cells | *Zikherman et al. (2012)*, Figure S1G; Manuscript *Figure 1—figure supplement 3A* |
| | Pam3CSK4 | TLR1/2 | Nur77-eGFP | B cells | Manuscript *Figure 1—figure supplement 3A* |
| | Anti-Igκ | BCR | Nur77-eGFP | B cells | Manuscript *Figure 2F* |
| Modulates pathway at steady state in vivo | IgHEL Tg | Antigen/ BCR | Nur77-eGFP | B cells | *Zikherman et al. (2012)*, *Figure 3B,C*; Manuscript *Figure 1A* |
| | IgHEL BCR Tg/sHEL Ag | Antigen/ BCR | Nur77-eGFP | B cells | *Zikherman et al. (2012)*, Figure 3B,C; Manuscript *Figure 1A* |
| | *Lyn*$^{-/-}$ | BCR via ITIMs | Nur77-eGFP | B cells | Manuscript *Figure 5—figure supplement 3A* |
| | CD45 allelic series | BCR via SFKs | Nur77-eGFP | B cells | *Zikherman et al. (2012)*, Figure 3A,B |

DOI: https://doi.org/10.7554/eLife.35074.010

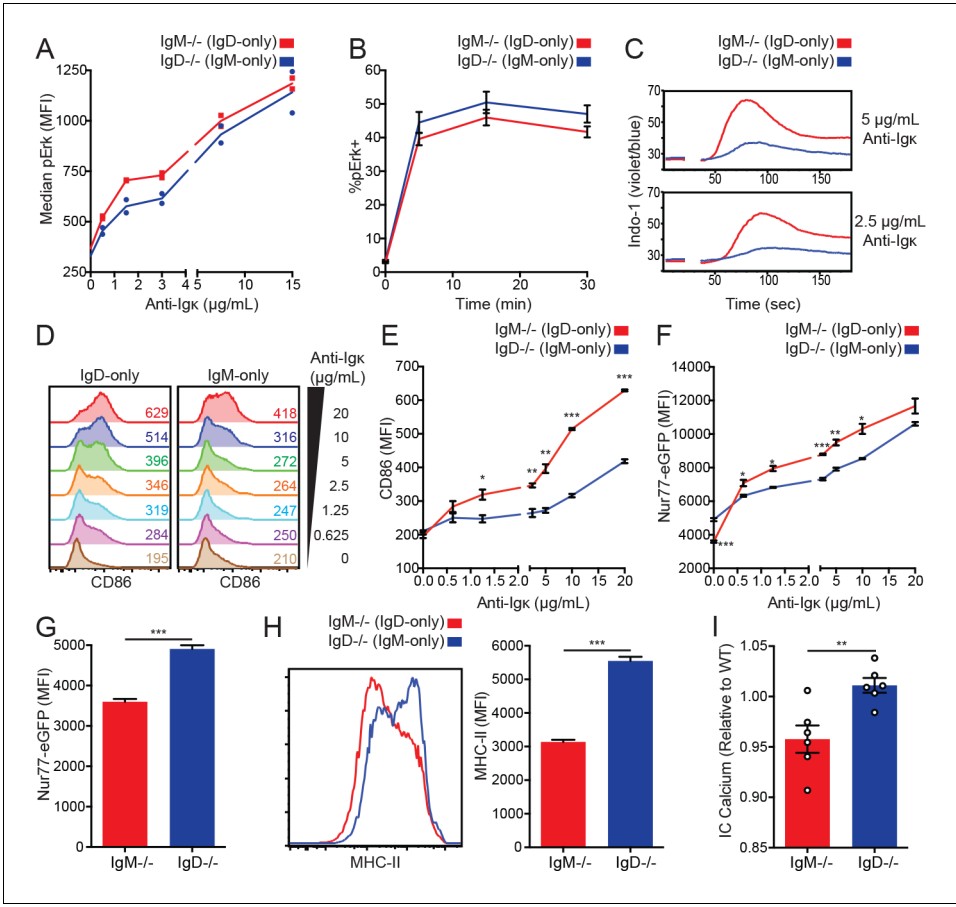

**Figure 2.** IgD signals strongly in vitro but weakly in vivo. (**A**) Median intracellular pErk in splenic CD23+ B cells stimulated with anti-Igκ for 15 min. (**B**) Erk phosphorylation kinetics in splenic CD23+ B cells stimulated with 15 μg/ mL anti-Igκ. (**C**) Splenocytes from $IgM^{-/-}$ and $IgD^{-/-}$ mice were loaded with Indo-1 and stimulated with 2.5 or 5 μg/mL anti-Igκ. B220+CD23+CD93- Fo B cells are compared. (**D**) CD86 induction in CD23+ $IgM^{-/-}$ and $IgD^{-/-}$ splenocytes stimulated with anti-Igκ for 18 hr. (**E**) Summary data for CD86 MFI in (**D**). (**F**) Nur77-eGFP induction in cells from (**D**). (**G**) Nur77-eGFP in cells from (**D**) incubated with medium alone (0 μg/mL anti-Igκ). (**H**) Representative histograms and summary data for MHC-II induction in unstimulated cells from (**D**). (**I**) Basal calcium in unstimulated $IgM^{-/-}$ and $IgD^{-/-}$ Fo B cells was calculated by normalizing the geometric mean of [Indo-1(violet)/ Indo-1(blue)] to WT B cells in the same experiment. For (**A**), signaling in cells from n = 2 $IgM^{-/-}$ and $IgD^{-/-}$ mice is displayed, and results for 1.5, 3, and 15 μg/mL of anti-Igκ were replicated in n = 3 independent experiments. Data in (**B**) was compiled from n = 3 independent experiments with n = 3 mice of each genotype in each experiment. Data in (**C**) are representative of n = 4 independent experiments for 5 μg/mL and n = 2 independent experiments for 2.5 μg/mL. For (**D-H**), values were calculated for splenocytes from n = 3 mice of each genotype. For (**I**), basal calcium ratios from n = 6 independent experiments are compiled. Welch's t test was used to calculate p values, and mean ±SEM is displayed. *p<0.05, **p<0.01, ***p<0.001.

DOI: https://doi.org/10.7554/eLife.35074.011

The following source data and figure supplement are available for figure 2:

**Source data 1.** Numerical data corresponding to *Figure 2A and B, E-I,* and *Figure 2—figure supplement 1A*.
DOI: https://doi.org/10.7554/eLife.35074.013
**Source data 2.** Numerical data corresponding to basal calcium in *Figure 2I*.
DOI: https://doi.org/10.7554/eLife.35074.014
**Figure supplement 1.** S6 and calcium signaling in $IgM^{-/-}$ and $IgD^{-/-}$ B cells.
DOI: https://doi.org/10.7554/eLife.35074.012

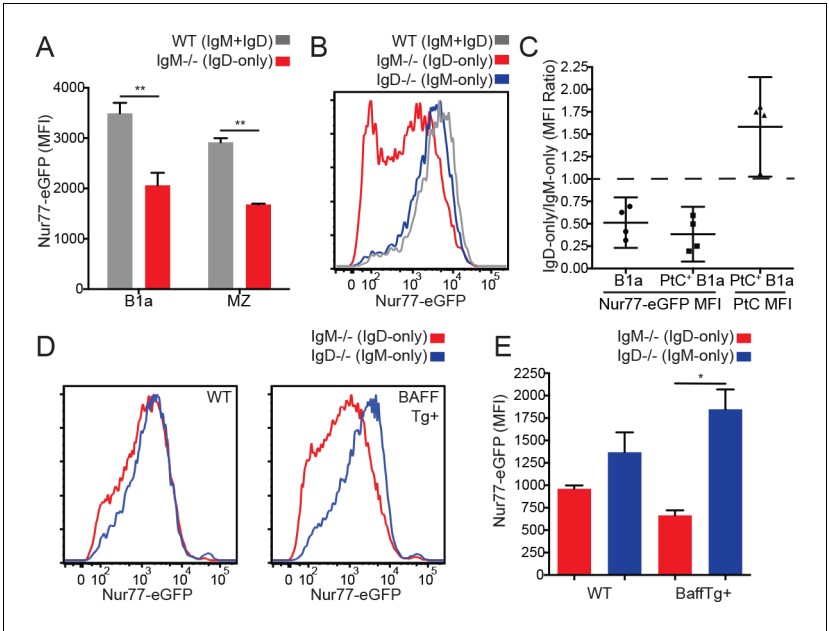

**Figure 3.** Reduced in vivo antigen sensing by IgD in innate-like B cells. (**A**) Nur77-eGFP in peritoneal B1a (CD19 +CD5+CD23-) and splenic MZ (B220+CD21$^{hi}$CD23$^{lo}$) B cells from WT and $IgM^{-/-}$ mice. (**B**) Nur77-eGFP in PtC-binding peritoneal B1a cells from WT, $IgM^{-/-}$, and $IgD^{-/-}$ mice. (**C**) Nur77-eGFP and PtC MFIs were calculated for total B1a and PtC-binding B1a cells. The ratio of the $IgM^{-/-}$ (IgD-only) MFI to the $IgD^{-/-}$ (IgM-only) MFI is displayed with a 95% confidence interval. (**D**) Representative histograms of Nur77-eGFP in $IgM^{-/-}$ and $IgD^{-/-}$ splenic MZ B cells from mice without (left) and with (right) a BAFF overexpression transgene. (**E**) Quantification of Nur77-eGFP in MZ B cells from (**D**). For (A) and (E), n = 3 mice of each genotype were analyzed. For (B), histograms are representative of n = 4 mice of each genotype. For (C), ratios are pooled for n = 4 mice of each genotype from three independent experiments. For (D), histograms are representative of n = 3 mice of each genotype. Welch's t test (A and E) was used to calculate p values, and mean +SEM is displayed (except in C). *p<0.05.

DOI: https://doi.org/10.7554/eLife.35074.015

The following source data and figure supplement are available for figure 3:

**Source data 1.** Numerical data corresponding to Nur77-eGFP and BCR expression in innate-like B cells in *Figures 3A, C, E, Figure 3—figure supplement 1A-C*.
DOI: https://doi.org/10.7554/eLife.35074.017

**Source data 2.** Numerical data corresponding to Nur77-eGFP expression and PtC binding in B1a cells in *Figure 3C*.
DOI: https://doi.org/10.7554/eLife.35074.018

**Figure supplement 1.** The effect of BAFF, competition, and allotype on Igκ and Nur77-eGFP expression.
DOI: https://doi.org/10.7554/eLife.35074.016

## Dual expression of IgM and IgD BCRs is required to establish a broad dynamic range of BCR responsiveness across the repertoire

Surface IgM, but not IgD, expression is inversely correlated with endogenous antigen recognition and GFP expression across a diverse repertoire of mature naïve follicular (Fo) B cells, such that B cells reactive to endogenous antigens express high levels of IgD and low levels of IgM on their surface (*Figure 1B*) (*Zikherman et al., 2012*). The variation in surface IgM expression across the B cell repertoire is profound, spanning a 100-fold range. However, because IgD is expressed at high and invariant levels in all WT naïve Fo B cells, total surface BCR, unlike IgM, has little dynamic range across the WT repertoire (*Figure 1—figure supplement 2A*).

To explore how IgM and IgD differentially regulate B cells reactive to endogenous antigens, we generated Nur77-eGFP mice deficient for either IgM or IgD (*Lutz et al., 1998*; *Nitschke et al., 1993*). In $IgM^{-/-}$ mice, all B cells express IgD alone prior to class switch recombination (CSR), and in $IgD^{-/-}$ mice, all B cells express IgM alone prior to CSR. We observed that IgM is still downregulated

on $IgD^{-/-}$ GFP$^{hi}$ Fo B cells, but the dynamic range of IgM expression is highly restricted relative to the broad range observed on WT cells (*Figure 1B*).

We previously showed that IgM down-modulation on GFP$^{hi}$ reporter B cells largely accounts for impaired signaling through IgM (*Figure 1C*) (*Zikherman et al., 2012*). However, the narrow dynamic range of IgM on IgD-deficient cells renders reporter B cells unable to fine-tune signaling through the IgM BCR across the GFP repertoire (*Figure 1C*). Furthermore, IgD-mediated signaling is not altered across the GFP repertoire in the presence or absence of IgM (*Figure 1C*). As a result, reduced responsiveness of GFP$^{hi}$ B cells is most profound in B cells that express the IgM isotype in the presence of IgD. This suggests that IgD expression may be necessary to establish and/or maintain a broad and functionally dynamic range of surface IgM expression across the B cell repertoire. This is consistent with a previously proposed survival function for the IgD BCR (*Roes and Rajewsky, 1993*; *Sabouri et al., 2016*).

## $IgM^{-/-}$ follicular B cells express more surface BCR than $IgD^{-/-}$ follicular B cells, but similar Nur77-eGFP

Deletion of either the IgM or the IgD BCR isotype results in compensatory upregulation of the remaining isotype. This may be due in part to a change in competition for pairing with Igα/β heterodimers, which is essential for trafficking of IgM to the cell surface (*Hombach et al., 1990*; *Sabouri et al., 2016*). $IgM^{-/-}$ B cells express about twice as much surface BCR as $IgD^{-/-}$ B cells, as measured by surface anti-light chain staining (*Figure 1D* and *Figure 1—figure supplement 2B–E*). Nevertheless, distribution of Nur77-eGFP expression across the naïve B cell repertoire is nearly identical in $IgM^{-/-}$ and $IgD^{-/-}$ Fo B cells (*Figure 1E*). Further, at every level of Nur77-eGFP, cells that express IgD alone require more surface BCR to drive an equivalent amount of GFP compared to cells that express IgM alone (*Figure 1F*). This suggests that on a per-receptor basis, IgD BCR may be less efficient at inducing Nur77-eGFP in response to endogenous antigens. Importantly, this is not attributable to differential dependence upon IgM or IgD expression for signaling via CXCR4 or downstream of canonical TLR ligands (*Figure 1—figure supplement 1C*, *Figure 1—figure supplement 3A*).

## IgD BCR crosslinking induces robust signaling in vitro

Because IgD drives less Nur77-eGFP per receptor than IgM in vivo, we wanted to determine whether there were defects in signal transduction downstream of IgD. To mimic antigen binding to the membrane-distal end of the BCR and allow for direct comparison between the isotypes, we stimulated IgD-only ($IgM^{-/-}$) and IgM-only ($IgD^{-/-}$) cells with anti-Igκ and compared downstream signaling. BCR ligation in IgD-only B cells induced equivalent amounts of Erk phosphorylation relative to IgM-only B cells (*Figure 2A*), and there were no gross differences in pErk kinetics (*Figure 2B*). Additionally, IgD-only cells induced more robust intracellular calcium increase and S6 phosphorylation than IgM-only cells (*Figure 2C* and *Figure 2—figure supplement 1A*). This was not due to differential effects of the Fc portion of the stimulatory antibody on IgM-only and IgD-only B cells as anti-Igκ-F(ab')$_2$ and anti-Igκ stimulation produced identical signaling (*Figure 2—figure supplement 1B*).

## IgD BCRs sense endogenous antigens less efficiently than IgM BCRs

Due in part to increased surface receptor expression, IgD-only B cells induce comparable Erk phosphorylation and enhanced calcium mobilization relative to IgM-only B cells stimulated with anti-Igκ. As a result, we observe significantly more CD86 and Nur77-eGFP induction in IgD-only B cells after 18 hr of anti-Igκ stimulation in vitro (*Figure 2D–F*). However, IgD-only cells induced less Nur77-eGFP and MHC-II upregulation when cultured in the absence of exogenous stimulus (*Figure 2G and H*). This discrepancy suggests that despite increased receptor expression and efficient coupling to downstream signaling machinery, residual endogenous antigens occupying the IgD BCR are less efficient at inducing signaling ex vivo. Consistent with this hypothesis, basal calcium analyzed immediately ex vivo is depressed in cells that express only IgD (*Figure 2I*). These data suggest that while signal transduction downstream of the IgD BCR is robust in vitro, IgD responds less efficiently than IgM to the relevant antigens it encounters in vivo.

## Reduced endogenous antigen sensing by IgD BCR in innate-like B cells

While IgD-only and IgM-only Fo B cells express comparable levels of Nur77-eGFP (*Figure 1E*), we suspected that competitive pressures for survival might constrain the acceptable range of BCR signaling among mature Fo B cells. We further speculated that compensatory mechanisms such as altered surface receptor and altered BCR repertoire might fine tune how much signaling Fo B cells experience in vivo in order to lie within this range. In contrast to Fo B cells, IgD-only marginal zone (MZ) and B1a cells expressed significantly less Nur77-eGFP than WT cells despite higher levels of surface BCR (*Figure 3A*, *Figure 1—figure supplement 2B–E*). To determine whether this difference was due to isotype or altered BCR repertoire, we probed GFP expression in B1a cells specific for phosphatidylcholine (PtC), an endogenous antigen exposed on the surface of dying cells that is thought to select developing B cells into the B1a compartment (*Baumgarth, 2011*). We found a large difference in GFP even among B1a cells with a common specificity, and this occurs in spite of high surface PtC binding in IgD-only B1a cells (*Figure 3B and C*).

We took an independent and complementary approach to address the same question in MZ B cells; we crossed $IgM^{-/-}$ and $IgD^{-/-}$ reporter mice to a genetic background that drives over-expression of the B cell survival factor BAFF (*Gavin et al., 2005*). BAFF overexpression 'unrestricts' the BCR repertoire by removing competition for survival, thereby reducing both positive and negative selection pressures and permitting cells with either insufficient or excessive BCR signaling to persist in the repertoire, resulting in a massively expanded MZ compartment (*Stadanlick and Cancro, 2008*). Importantly, BAFF does not directly regulate reporter GFP expression, and therefore GFP reflects endogenous antigen stimulation on this genetic background (*Zikherman et al., 2012*). In BAFF Tg mice, IgD-only MZ cells expressed much less GFP than those expressing IgM alone (*Figure 3D and E*). This was not explained by differences in surface receptor levels between these populations as they are comparable on this genetic background (*Figure 3—figure supplement 1A*). Indeed, IgD-only cells with low GFP and IgM-only cells with high GFP appear to be preferentially rescued in mice with excess BAFF, suggesting that distinct BCR repertoires, far from accounting for GFP differences between IgM-only and IgD-only MZ B cells, actually obscure these differences. Conversely, either 'fixing' or unrestricting the BCR repertoire unmasks impaired GFP upregulation by IgD relative to IgM.

As a result of allelic exclusion of the heavy chain locus during B cell development, $IgM^{+/-}$ heterozygous mice develop two genetically distinct sets of B cells in which half express only IgD and the other half express both IgM and IgD. We exploited this property to confirm that the Nur77-eGFP difference in the MZ and B1a compartments is cell intrinsic and not a consequence of the presence or absence of serum IgM (*Figure 3—figure supplement 1B*).

Since $IgM^{-/-}$ and $IgD^{-/-}$ mice were generated on the Balb/c and 129 genetic backgrounds respectively, their germline VDJ loci are different from that of C57BL/6 mice to which they have been back-crossed. Throughout our study, we have validated our findings in either Balb/c-B6 F1 mice or $IgH^{a/b}$ heterozygous mice, which have two sets B cells with identical IgM and IgD isotype expression but distinct germline VDJ loci. We took this approach to confirm that differences in MZ GFP were due to BCR isotype and not VDJ locus (*Figure 3—figure supplement 1C*).

## Cell-intrinsic skewing of B cell development by the IgM and IgD BCRs

Generation of the B1a compartment requires endogenous antigen recognition and strong BCR signaling, while the opposite is true for MZ B cells (*Figure 4—figure supplement 1A*) (*Cariappa and Pillai, 2002*; *Casola et al., 2004*). Indeed, B1a cells are modestly reduced in IgM-deficient mice, while MZ B cells are modestly increased (*Figure 4—figure supplement 1B–C*). Because serum IgM deficiency leads to increased B1a numbers, we assessed B1a and MZ B cell development in a competitive setting to isolate the cell-intrinsic effects of IgM and IgD BCRs (*Boes et al., 1998*). Analogous to the $IgM^{+/-}$ mice described above, $IgM^{+/-}$ $IgD^{-/+}$ heterozygous mice develop two genetically distinct sets of B cells in which half express only IgD and the other half express only IgM (*Figure 4A*). Although IgD-only B cells can partially populate the B1a compartment in the absence of competition (*Lutz et al., 1998*; *Übelhart et al., 2015*), IgD-only cells are virtually excluded from this compartment in a competitive setting (*Figure 4B*). Indeed, this parallels the loss of PtC-binding IgD-only peritoneal B cells observed in competition with wild type cells (*Übelhart et al., 2015*). In contrast, IgD-only B cells preferentially populate the MZ B cell compartment in competition with

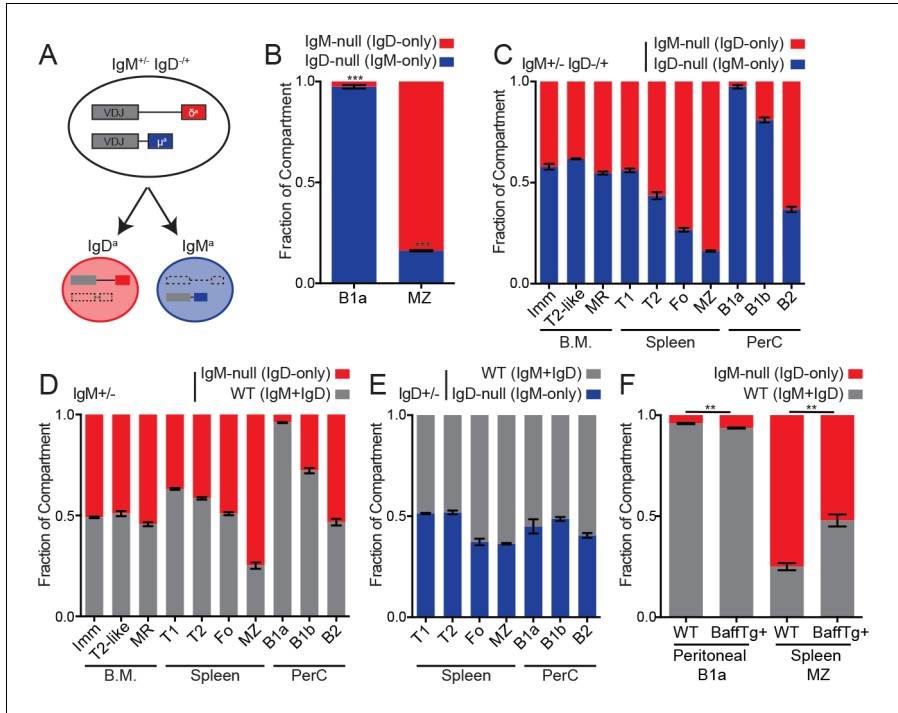

**Figure 4.** Cell-intrinsic skewing of B cell development by IgM and IgD BCRs. (**A**) Allelic exclusion leads to a 1:1 mixture of IgM-only and IgD-only B cells in $IgM^{+/-}\ IgD^{-/+}$ mice. (**B**) Proportion of peritoneal B1a (CD19+CD5+CD23-) and splenic MZ (B220+CD21hiCD23lo) B cells originating from each Ig locus in $IgM^{+/-}\ IgD^{-/+}$ mice. (**C**) Relative competition between IgM+ and IgD+ B cells in $IgM^{+/-}\ IgD^{-/+}$ mice was calculated for bone marrow, splenic, and peritoneal B cell compartments. Results include data from (**B**) for reference. Immature (CD23-CD93+); T2-like (CD23+CD93+); mature recirculating (CD23+CD93-); T1 (CD93+CD23-); T2/3 (CD93+CD23+); Fo (CD93-CD23+); MZ (CD21hiCD23lo); B1a (CD5+CD23-); B1b (CD5-CD23-); B2 (CD5-CD23+). (**D**) Relative competition between WT (IgMb+) and IgM-null (IgDa+) B cells in $IgM^{+/-}$ mice was determined as in (**C**). (**E**) Relative competition between WT (IgMb+) and IgD-null (IgMa+) B cells in $IgD^{+/-}$ mice was determined for splenic and peritoneal compartments as described in (**C**). (**F**) Competition in peritoneal B1a and splenic MZ compartments in $IgM^{+/-}$ mice with or without a BAFF overexpression transgene. Results include data from (**D**) for reference. For (B) and (C), n = 3–5 mice were analyzed. For (D), n = 3–8 mice were analyzed. For (E), n = 5 mice were analyzed. For (F), n = 4–5 mice of each genotype were analyzed. Welch's t test was used to calculate p values, and mean ±SEM is displayed. **p<0.01, ***p<0.001.

DOI: https://doi.org/10.7554/eLife.35074.019

The following source data and figure supplement are available for figure 4:

**Source data 1.** Numerical data corresponding to competition between IgM-only, IgD-only and WT B cells in *Figure 4B–F,* and in *Figure 4—figure supplement 1B-D*
DOI: https://doi.org/10.7554/eLife.35074.021

**Source data 2.** Numerical data corresponding to competition between IgM-only, IgD-only and WT B cells in *Figure 4B–E*.
DOI: https://doi.org/10.7554/eLife.35074.022

**Source data 3.** Numerical data corresponding to competition between IgD-only and WT B cells in *Figure 4F*.
DOI: https://doi.org/10.7554/eLife.35074.023

**Figure supplement 1.** B cell subset development in $IgM^{-/-}$, $IgD^{-/-}$, WT, and $IgH^{a/b}$ mice.
DOI: https://doi.org/10.7554/eLife.35074.020

IgM-only B cells (*Figure 4B*). These data are consistent with a signal strength model of B1a/MZ B cell development (*Figure 4—figure supplement 1A*), and this suggests that in vivo signaling, antigen recognition, or both are reduced in IgD-only B cells. These differences in signaling could in turn modulate the generation and/or survival of IgD-only B1a and MZ B cells.

We sought to determine whether skewed development of B cell lineages was driven primarily by properties of IgD, IgM, or both by comparing development of B cell populations in $IgM^{+/-}\ IgD^{-/+}$,

*IgM*<sup>+/−</sup>, and *IgD*<sup>+/−</sup> mice (*Figure 4C–4E*). We conclude that skewing of MZ and B1a fates is primarily attributable to lack of IgM expression on IgD-only cells rather than lack of IgD expression on IgM-only cells because we observe little skewing in *IgD*<sup>+/−</sup> mice (*Figure 4E*). Further, the competitive advantage of IgD-only cells in the marginal zone niche is reduced by BAFF over-expression, consistent with loosening of competitive selection pressures (*Figure 4F*).

IgM-only B cells exhibit a disadvantage in the mature Fo compartment relative to IgD-only B cells (*Figure 4C*). We find that absence of IgD rather than excess IgM expression accounts for some of this disadvantage as WT and IgD-only cells compete equally well in this compartment (*Figure 4D*). This may reflect impaired positive selection or survival in absence of IgD. Indeed, similar to our observations, a disadvantage for Fo B cells lacking IgD was observed in *IgD*<sup>+/−</sup> mice previously (*Roes and Rajewsky, 1993*). As with differences in Nur77-eGFP expression, we confirmed that the developmental fates of IgM-only and IgD-only cells were due to BCR isotype and not VDJ locus (*Figure 4—figure supplement 1D*).

## Either IgD or IgM BCR is sufficient to mediate polyclonal B cell activation and germinal center differentiation in *Lyn*<sup>−/−</sup> mice

Mice deficient for either IgM or IgD can mount T-independent and T-dependent immune responses to model antigens (*Lutz et al., 1998*; *Nitschke et al., 1993*; *Roes and Rajewsky, 1993*). However, endogenous antigens may have unique properties that are absent in model antigen systems. To test whether each BCR isotype was competent to mediate autoimmune responses to endogenous antigens in lupus-prone mice, we generated *IgM*<sup>+/−</sup> and *IgD*<sup>+/−</sup> mice on the *Lyn*<sup>−/−</sup> background (*Chan et al., 1997*). The Src family kinase Lyn is essential to mediate ITIM-dependent inhibitory signals in B cells and myeloid cells (*Scapini et al., 2009*; *Xu et al., 2005*). *Lyn*<sup>−/−</sup> mice consequently develop a spontaneous lupus-like disease characterized by anti-DNA antibodies and nephritis on the C57BL/6 genetic background (*Chan et al., 1997*; *Hibbs et al., 1995*; *Nishizumi et al., 1995*). It has been shown that both B-cell-specific MyD88 expression and T cells are essential for IgG2a/c anti-dsDNA autoantibody production in *Lyn*<sup>−/−</sup> mice, and conditional deletion of Lyn in B cells is sufficient for autoimmunity (*Hua et al., 2014*; *Lamagna et al., 2014*).

Secreted natural IgM is thought to play important homeostatic functions that repress autoimmunity, particularly clearance of dead cell debris (*Boes et al., 2000*; *Manson et al., 2005*). By performing our analysis in mice with wild type and IgM-only or IgD-only B cells in a common milieu (*IgM*<sup>+/−</sup> *Lyn*<sup>−/−</sup> and *IgD*<sup>+/−</sup> *Lyn*<sup>−/−</sup> mice), we could control for this and isolate the B cell-intrinsic effects of the IgM and IgD BCRs. Importantly, precursor Fo B cells are present at roughly equal ratios from WT and IgM-only or IgD-only loci in these mice, and this is unaffected by VDJ background (*Figure 5—figure supplement 1A–C*). Moreover, *Lyn*<sup>−/−</sup> B cells expressing either IgM or IgD alone exhibit enhanced BCR signaling in vitro relative to *Lyn*<sup>+/+</sup> B cells, suggesting that neither isotype is uniquely coupled to ITIM-containing receptors (*Figure 5—figure supplement 2A–B*).

Polyclonal B cell activation in *Lyn*<sup>−/−</sup> mice precedes and is genetically separable from autoantibody production; it is driven by B cell-intrinsic loss of Lyn, is thought to result from enhanced BCR signaling, and is independent of MyD88 expression (*Hua et al., 2014*; *Lamagna et al., 2014*). Indeed, *Lyn*<sup>−/−</sup> reporter B cells have elevated Nur77-eGFP expression consistent with enhanced BCR signal transduction in vivo (*Figure 5—figure supplement 3A*). We had expected that activation of IgD-only *Lyn*<sup>−/−</sup> B cells might be impaired due to reduced sensing of endogenous antigens, but neither CD86 nor CD69 upregulation were significantly different on WT and IgD-only B cells in the absence of Lyn (*Figure 5A*). This implies that both BCRs can provide sufficient signals to drive polyclonal activation by endogenous antigens.

Although GC fate is disfavored relative to SLPC fate in *Lyn*<sup>−/−</sup> mice, GCs do arise spontaneously with time (*Hibbs et al., 1995*; *Hua et al., 2014*). We found that both IgD-only and IgM-only cells in Lyn-deficient mice made comparable contributions to the GC compartment in competition with wild type B cells, independent of VDJ background (*Figure 5B* and *Figure 5—figure supplement 3B–C*). We propose that endogenous antigen sensing by IgD, while normally dampened, is sufficient to drive both polyclonal activation and GC differentiation on the *Lyn*<sup>−/−</sup> background. This further implies that antigen capture by IgD and presentation on MHC-II is robust enough to recruit adequate T cell help to support these cell fates.

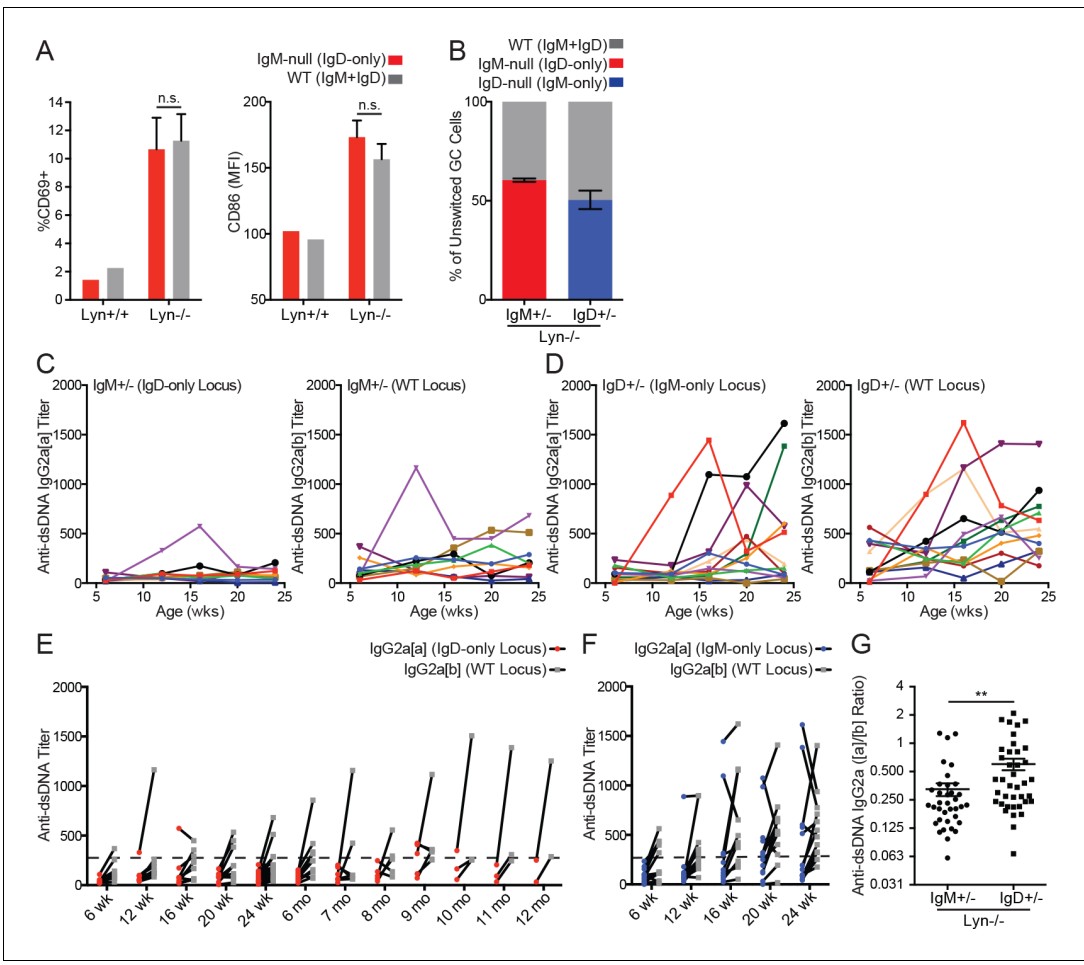

**Figure 5.** IgD can drive polyclonal activation and germinal center entry, but not anti-dsDNA IgG2a production, in $Lyn^{-/-}$ mice. (**A**) Surface CD69 and CD86 expression on CD23+ splenic B cells from each Ig locus in $IgM^{+/-}$ mice on $Lyn^{+/+}$ and $Lyn^{-/-}$ backgrounds. (**B**) Percentage of unswitched germinal center (CD19+ Fas[hi] GL-7[hi] IgM/IgD+) B cells from each Ig locus in $IgM^{+/-}$ and $IgD^{+/-}$ mice on the $Lyn^{-/-}$ background. (**C**) Anti-dsDNA IgG2a titers from each Ig locus in $IgM^{+/-}$ $Lyn^{-/-}$ mice were calculated by ELISA using pooled $IgH^{a/b}$ autoimmune serum with high-titer autoantibodies from each locus as a reference (titer set at 1000). Each color represents a single mouse tracked over time. (**D**) Anti-dsDNA IgG2a titers in $IgD^{+/-}$ $Lyn^{-/-}$ mice were calculated as in (**C**). (**E**) Paired anti-dsDNA titers from each locus in individual $IgM^{+/-}$ $Lyn^{-/-}$ mice from (**C**) with additional mice from 24 weeks to 12 months. (**F**) Paired anti-dsDNA titers from each locus in $IgD^{+/-}$ $Lyn^{-/-}$ mice from (**D**). (**G**) Ratio of anti-dsDNA IgG2a[a] to IgG2a[b] from all samples in (**E**) and (**F**) with an anti-dsDNA IgG2a titer >250 from either locus; cutoff defined by titers in young WT mice. For (**A**), n = 4 $IgM^{+/-}$ $Lyn^{-/-}$ mice are compared to a reference $IgM^{+/-}$ $Lyn^{+/+}$ mouse. Qualitatively similar results were obtained in two independent experiments. For (**B**) n = 5–6 mice of each genotype were analyzed. For (**C**) and (**D**), n = 9 and n = 12 mice were tracked. For (**G**), n = 36 $IgM^{+/-}$ $Lyn^{-/-}$ and n = 39 $IgD^{+/-}$ $Lyn^{-/-}$ anti-dsDNA IgG2a+ samples were compared. Welch's t test was used to calculate p values, and mean ±SEM is displayed. **p<0.01.

DOI: https://doi.org/10.7554/eLife.35074.024

The following source data and figure supplements are available for figure 5:

**Source data 1.** Numerical data corresponding to *Figure 5A-G*, *Figure 5—figure supplement 1B-C*, *3C-E*.
DOI: https://doi.org/10.7554/eLife.35074.028
**Figure supplement 1.** Splenic B cell subsets in $IgH^{a/b}$, $IgM^{+/-}$, and $IgD^{+/-}$ $Lyn^{-/-}$ mice.
DOI: https://doi.org/10.7554/eLife.35074.025
**Figure supplement 2.** Signaling in $IgM^{+/-}$ and $IgD^{+/-}$ $Lyn^{-/-}$ B cells.
DOI: https://doi.org/10.7554/eLife.35074.026
**Figure supplement 3.** Nur77-eGFP in $Lyn^{-/-}$ B cells and the role of BCR allotype in $Lyn^{-/-}$ phenotypes.
DOI: https://doi.org/10.7554/eLife.35074.027

## IgM BCR is required to drive dsDNA antibody production in $Lyn^{-/-}$ mice

IgM- and IgD-deficient B cells express BCR heavy chain allotype [a] (IgH[a]), and their secreted antibodies can be differentiated from WT B6 antibodies (IgH[b]) even after isotype switching (e.g. IgG2a [a] vs. IgG2a[b] – also referred to as IgG2c). Allotype-specific antibodies have been previously used in the context of lupus mouse models to track autoantibodies generated by B cells of distinct genetic origin (**Mills et al., 2015**; **Pisitkun et al., 2006**). We took an analogous approach to assess IgG2a/c anti-dsDNA autoantibodies emanating from each genetic locus in $IgM^{+/-}$, $IgD^{+/-}$, and $IgH^{a/b}$ control mice deficient for Lyn. In order to compensate for reduced penetrance of autoantibody production in $IgM^{+/-} Lyn^{-/-}$ mice, we collected serum from this genotype at time points beyond 24 weeks to increase the number of samples in which tolerance had been broken. While B cells expressing IgM (of either IgH locus) generated IgG2a/c anti-dsDNA, IgD-only B cells were relatively protected from generating anti-dsDNA antibodies, even at late time points (**Figure 5C–G**, **Figure 5—figure supplement 3D–E**). Importantly, this is attributable to BCR isotype and not VDJ locus (**Figure 5—figure supplement 3D–E**).

We noted that anti-dsDNA antibody titers fluctuated substantially over time in $Lyn^{-/-}$ mice, in some cases dropping markedly over a four-week period (**Figure 5C and D**). Anti-dsDNA IgG autoantibodies in patients with systemic lupus erythematosus (SLE) similarly fluctuate over time – in contrast to other anti-nuclear specificities - and correlate with disease flares (**Liu et al., 2011**). Furthermore, unmutated germline BCRs from naïve B cells are recruited directly into the circulating plasmablast compartment during SLE flares (**Tipton et al., 2015**). Taken together, this suggests that anti-dsDNA antibodies in mice and humans are secreted at least in part by SLPCs emerging from an extra-follicular immune response, rather than long-lived plasma cells (LLPCs) that originate in GCs. Our data suggests that, in the absence of Lyn, the IgD BCR is sufficient to mediate B cell activation and GC entry, but the IgM BCR is essential to produce dsDNA-specific SLPCs.

## IgM BCR is required for expansion of unswitched plasma cells in $Lyn^{-/-}$ mice

In addition to stochastic generation of autoantibodies over time, $Lyn^{-/-}$ mice exhibit a massive expansion of the splenic IgM plasma cell compartment that corresponds to a roughly 10-fold increase in serum IgM (**Hibbs et al., 1995**). While natural IgM is thought to be secreted by B1a-derived plasma cells (**Baumgarth, 2011**), elevated serum IgM in $Lyn^{-/-}$ mice arises from the Fo B2 compartment since $Lyn^{-/-}$ mice lack MZ B cells, and bone marrow chimeras lacking the B1a compartment reconstitute the hyper-IgM secretion phenotype (**Luo et al., 2014**). Moreover, while BCR signaling is prominently enhanced in $Lyn^{-/-}$ B2 B cells, it is markedly dampened in $Lyn^{-/-}$ B1a cells (**Figure 6—figure supplement 1A**) (**Skrzypczynska et al., 2016**).

IgM plasma cell expansion is temporally and genetically separable from autoimmunity in $Lyn^{-/-}$ mice; this phenotype develops in mice as young as 7–10 weeks of age with 100% penetrance, is B-cell intrinsic, and is not dependent upon MyD88 expression (**Hua et al., 2014**; **Infantino et al., 2014**; **Lamagna et al., 2014**). A pathway involving Btk, Ets1, and Blimp-1 is thought to drive this phenotype. Ets1 inhibits plasma cell differentiation in part by antagonizing the key transcriptional regulator of plasma cell fate, Blimp-1 (**John et al., 2008**). Mice deficient for Ets1 display IgM plasma cell expansion, Ets1 levels are reduced in $Lyn^{-/-}$ mice, and restoration of Ets1 expression normalizes IgM plasma cell numbers (**Luo et al., 2014**). Furthermore, reducing levels of Btk, a critical kinase that mediates BCR signaling, is sufficient to suppress IgM plasma cell expansion in $Lyn^{-/-}$ mice, and Ets1 levels are concomitantly restored to normal levels in $Btk^{lo} Lyn^{-/-}$ mice (**Gutierrez et al., 2010**; **Luo et al., 2014**; **Mayeux et al., 2015**). Therefore, IgM plasma cell expansion in $Lyn^{-/-}$ mice is driven by exaggerated Btk-dependent BCR stimulation of B2 B cells by endogenous antigens (**Figure 6—figure supplement 2A**). Indeed, enhanced BCR signaling has been shown to favor SLPC fate over GC differentiation in Lyn-sufficient mice as well (**Chan et al., 2009**; **Nutt et al., 2015**; **Paus et al., 2006**). We therefore wondered whether impaired endogenous antigen sensing by IgD influenced Ets1 expression and unswitched PC expansion in $Lyn^{-/-}$ mice.

As previously shown, $Lyn^{-/-}$ B cells downregulate Ets1 protein expression (**Figure 6A**) (**Luo et al., 2014**). However, Lyn-deficient B cells expressing only IgD did not do so, consistent with a reduced ability of IgD to transmit antigen-dependent signals in vivo (**Figure 6A**). As previously reported, we

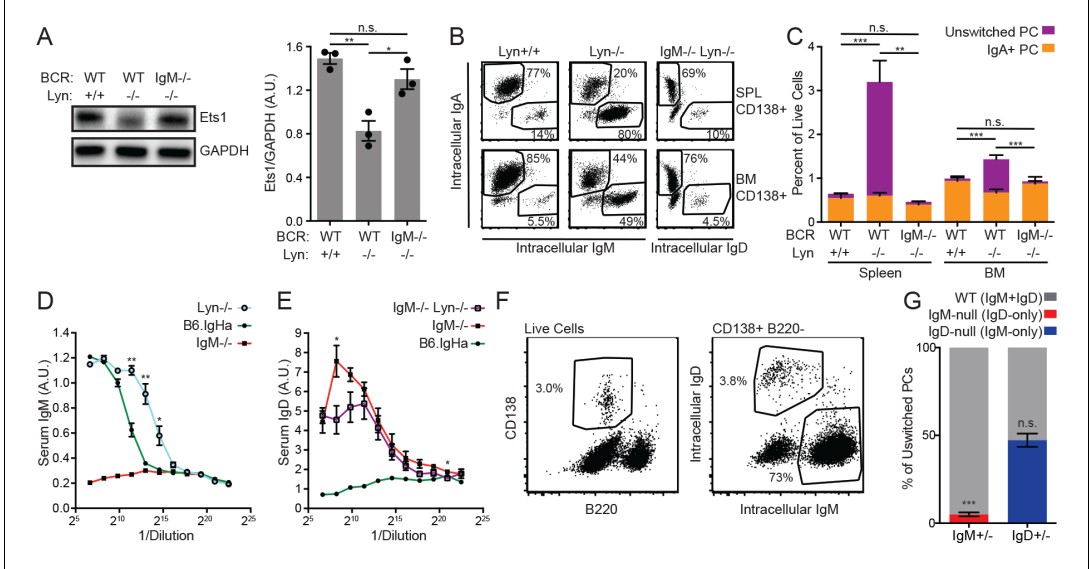

**Figure 6.** Cell-intrinsic IgM expression is required for unswitched plasma cell expansion in $Lyn^{-/-}$ mice. (A) Representative blot and quantification of Ets1 and GAPDH protein in purified splenic B cells from WT, $Lyn^{-/-}$, and $IgM^{-/-}$ $Lyn^{-/-}$ mice. (B) Composition of the CD138+ plasma cell compartments in the spleen and bone marrow of WT, $Lyn^{-/-}$, and $IgM^{-/-}$ $Lyn^{-/-}$ mice was determined by intracellular staining of IgM, IgD, and IgA. (C) Percentages in (B) multiplied by the fraction of live cells positive for CD138 in each tissue. Unswitched cells are positive for either IgM or IgD. Statistics correspond to unswitched plasma cell percentages; differences in IgA+ cells were not significant. (D) Serum IgM in 16-week-old mice was quantified for B6.IgHa (WT) and $Lyn^{-/-}$ mice by ELISA. A sample from an $IgM^{-/-}$ mouse is shown for reference. (E) Serum IgD in 16-week-old mice was quantified for $IgM^{-/-}$ and $IgM^{-/-}$ $Lyn^{-/-}$ mice by ELISA. A sample from a WT mouse is shown for reference. (F) Gating scheme for quantifying the unswitched splenic plasma cell composition of $IgM^{+/-}$ $Lyn^{-/-}$ mice. (G) Percentage of unswitched splenic plasma cells (CD138+B220$^{lo}$IgM/IgD+) from each locus in $IgM^{+/-}$ and $IgD^{+/-}$ mice on the $Lyn^{-/-}$ background. For (B) and (C), figures are representative of n = 4–5 mice of each genotype. For (D), values from n = 3 WT and n = 4 $Lyn^{-/-}$ mice are averaged. For (E), values from n = 3 $IgM^{-/-}$ and n = 4 $IgM^{-/-}$ $Lyn^{-/-}$ mice are averaged. For (F) and (G), n = 4–5 mice of each genotype were used. Welch's t test was used to calculate p values, and mean ±SEM is displayed. *p<0.05, **p<0.01, ***p<0.001.

DOI: https://doi.org/10.7554/eLife.35074.029

The following source data and figure supplements are available for figure 6:

**Source data 1.** Numerical data corresponding to Ets1 expression, plasma cell compartments, and serum IgM and IgD titers in *Figure 6A-G*.
DOI: https://doi.org/10.7554/eLife.35074.032
**Source data 2.** Numerical data corresponding to Ets1 expression in splenic B cells in *Figure 6A*.
DOI: https://doi.org/10.7554/eLife.35074.033
**Source data 3.** Numerical data corresponding to plasma cell compartments in *Figure 6B–C*.
DOI: https://doi.org/10.7554/eLife.35074.034
**Figure supplement 1.** BCR signaling in $Lyn^{-/-}$ peritoneal B cell subsets.
DOI: https://doi.org/10.7554/eLife.35074.030
**Figure supplement 2.** Lyn restrains unswitched plasma cell differentiation of follicular B cells.
DOI: https://doi.org/10.7554/eLife.35074.031

found that the frequency of plasma cells in the spleens of $Lyn^{-/-}$ mice was increased fourfold relative to $Lyn^{+/+}$ mice, but the bone marrow plasma cell compartment size was relatively unaffected (*Figure 6B–C*) (*Infantino et al., 2014*). The increase in splenic plasma cells was attributable entirely to unswitched (IgM$^+$) plasma cells (*Figure 6C*). However, unswitched plasma cell expansion was completely absent in $IgM^{-/-}$ $Lyn^{-/-}$ spleens (*Figure 6C*). This was reflected in steady-state serum antibody levels; while $Lyn^{-/-}$ mice have elevated serum IgM relative to WT mice, $IgM^{-/-}$ $Lyn^{-/-}$ mice did not have elevated serum IgD relative to $IgM^{-/-}$ mice (*Figure 6D–E*).

To confirm that resistance to unswitched plasma cell expansion was a cell-intrinsic feature of IgD-only $Lyn^{-/-}$ B cells, we assessed this phenotype in $IgM^{+/-}$ $Lyn^{-/-}$ mice. Similar to the non-competitive setting, IgD-only $Lyn^{-/-}$ cells did not contribute to an expansion in the unswitched plasma cell compartment (*Figure 6F-G*). This effect was due to isotype and not VDJ allele usage because both

IgH[a] and IgH[b] B cells in *IgH[a/b] Lyn[-/-]* mice generated unswitched plasma cells in a competitive setting (*Figure 5—figure supplement 3C*).

## IgM BCR is required for efficient generation of short-lived IgG1[+] plasma cells but is dispensable for GC B cell fate

IgD-only *Lyn[−/−]* Fo B cells are completely prevented from generating an expanded unswitched SLPC compartment in response to chronic endogenous antigen stimulation, but they are competent to enter the GC. To determine how this defect relates to responses towards exogenous model antigens, we sought to determine whether IgD-only *Lyn[+/+]* B cells exhibit skewing of cell fate in the context of a T-dependent response to a model antigen. To generate a robust polyclonal B cell response in which T cell help should not be limiting, we used the classic immunogen sheep RBCs. During this immune response, Fo B cells can either enter the germinal center or rapidly differentiate into IgG1[+]-SLPCs (*Chan et al., 2009*; *Paus et al., 2006*). We found that wild type, *IgM[−/−]*, and *IgD[−/−]* mice all produced extremely robust GC responses 5 days after SRBC injection, but B cells expressing IgD alone were unable to produce IgG1[+] SLPCs at this time point (*Figure 7A–D*).

Since serum IgM is absent in *IgM[−/−]* mice and is known to enhance PC generation (*Boes et al., 1998*), we wanted to assess the cell-intrinsic effect of IgD and IgM BCRs on the immune response to SRBCs. To do so, we immunized *IgM[+/−]* mice in which half of Fo B cells express IgD alone and half express both IgM and IgD. We could track the relative contribution of these two cell populations to the GC response because the bulk of GC B cells have not yet isotype-switched and express surface IgM or IgD. Both cell types made a robust contribution to the germinal center response (*Figure 7E*). Next we tracked the generation of IgG1[+] PCs by detecting allotype [a] or allotype [b] IgG1. In contrast to the GC response, IgD-only B cells were significantly disfavored in this compartment, although the defect was less severe than that observed in *IgM[−/−]* mice, which lack serum IgM (*Figure 7F*). To control for differences in the VDJ locus, we performed this experiment using *IgH[a/b]* control mice and observed no significant differences between allotypes, implying that BCR isotypes rather than VDJ locus accounted for our observations (*Figure 7—figure supplement 1A–B*).

In contrast to impaired IgG1 SLPC responses by IgD-only B cells, we observed that IgD-only B cells were able to generate unswitched plasma cells in response to SRBC immunization in both non-competitive and competitive settings (*Figure 7—figure supplement 1C–D*). Unswitched PC numbers correlate well with MZ B cell numbers in these mice (*Figure 4—figure supplements 1B* and *Figure 4D*). Moreover, MZ B cells efficiently generate short-lived unswitched plasma cells even in response to TD-antigens (*Phan et al., 2005*; *Song and Cerny, 2003*). We therefore suspect that IgG1 PCs emanate from the Fo B cell compartment, while the unswitched PC response may originate in the MZ B cell compartment. Normal SLPC generation by IgM-deficient MZ B cells in response to immunization would be consistent with previously reported normal T-independent II responses in IgM-deficient mice (*Lutz et al., 1998*). However, we cannot exclude a contribution of follicular-derived PCs that have failed to class switch to IgG1.

We sought to confirm that the IgG1[+] PC defect was generalizable to other T-dependent immunogens. The B cell response to NP hapten is stereotyped and makes predominant use of the VH186.2 heavy chain (*Loh et al., 1983*). We therefore immunized both *IgM[+/−]* and *IgH[a/b]* mice with NP hapten conjugated to rabbit serum albumin (NP-RSA) in order to control for differences in VDJ locus. Although allotype [a] makes a less robust response to NP than allotype [b], we normalized responses of IgD-only cells to wild type allotype [a] cells and found that GC responses at early time points were intact, but IgG1[+] PCs and NP-specific IgG1 titers were significantly reduced 7–8 days after immunization (*Figure 7G–I*). These data suggest that IgD-only follicular B cells are competent to enter the GC, but they have defects in adopting the SLPC fate (or class-switching) in response to T-dependent immunization.

## Discussion

We and others previously hypothesized that a major tolerance strategy employed by autoreactive Fo B cells is selective downregulation of IgM. However, since all Fo B cells express a high and invariant amount of surface IgD, it was unclear whether and how loss of IgM expression could affect B cell responses to antigenic stimulation. It was recently reported that IgM, but not IgD, is uniquely responsive to monomeric antigens due to a short and inflexible linker region coupling the Fab and

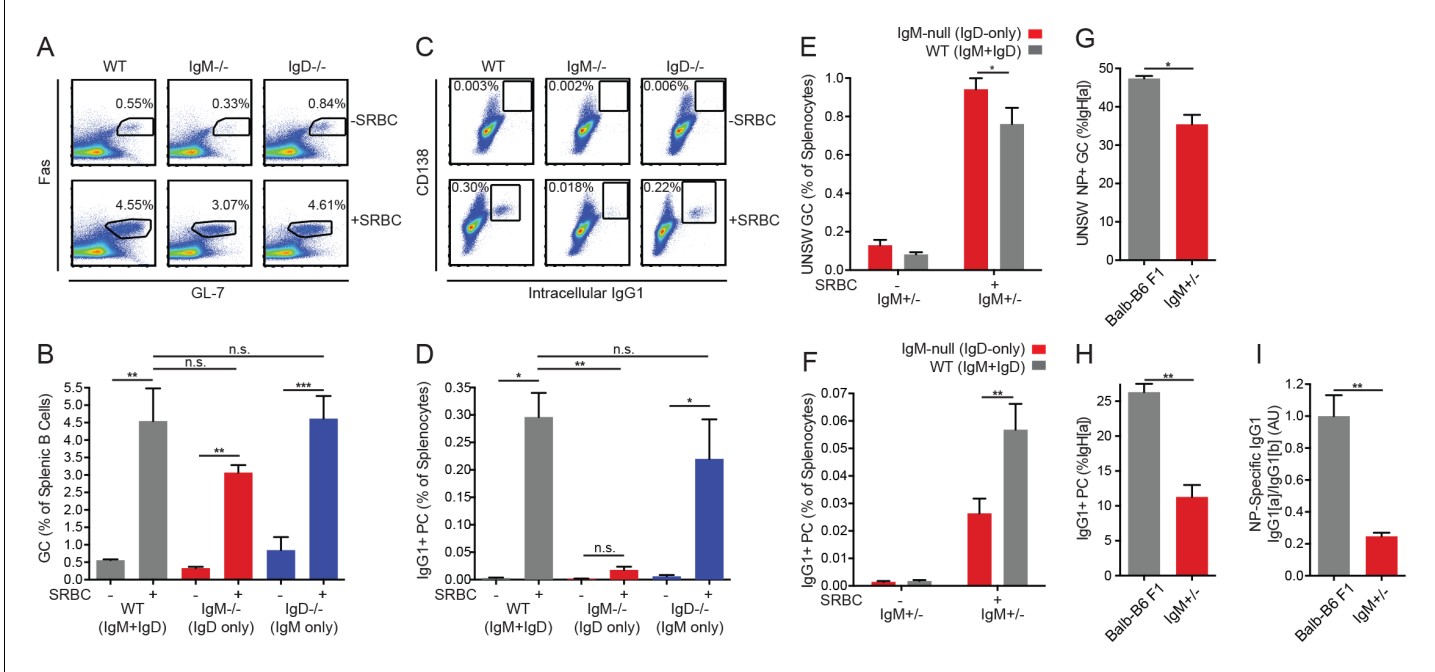

**Figure 7.** IgD-only cells have intact germinal center responses but impaired IgG1+ SLPC responses. (**A**) Splenic (CD19+) B cells from WT, $IgM^{-/-}$, and $IgD^{-/-}$ mice unimmunized or 5 days after i.p. immunization with 200 µL of 10% SRBCs. (**B**) Quantification of germinal center (Fas$^{hi}$ GL-7$^{hi}$) cells in (**A**). (**C**) Splenocytes from mice in (**A**). (**D**) Quantification of CD138+ IgG1+ plasma cells in (**C**). (**E**) WT (IgM$^b$+) and IgM-null (IgD$^a$+) germinal center B cells as a percentage of live splenocytes in unimmunized and $IgM^{+/-}$ mice 5 days after i.p. immunization with 200 µL of 10% SRBCs. (**F**) WT (IgG1$^b$+) and IgM-null (IgG1$^a$+) switched plasma cells (CD138 +IgG1+) as a percentage of live splenocytes in $IgM^{+/-}$ mice unimmunized or 5 days after i.p. immunization with 200 µL of 10% SRBCs. (**G**) Fraction of unswitched NP-specific germinal center cells (CD19+ Fas$^{hi}$ GL-7$^{hi}$ IgM/IgD+) from the IgH$^a$ locus in the spleens of Balb/c-B6 F1 and $IgM^{+/-}$ mice 7–8 days after i.p. immunization with 100 µg NP-RSA. (**H**) Fraction of IgG1+CD138+ plasma cells from the IgH$^a$ locus in Balb/c-B6 F1 and $IgM^{+/-}$ mice 7–8 days after i.p. immunization with 100 µg NP-RSA. (**I**) NP-specific IgG1$^a$ and IgG1$^b$ titers at OD = 0.2 were calculated for the mice in (**G–H**) by ELISA. The IgG1$^a$ to IgG1$^b$ titer ratio was calculated for each mouse, and all ratios were normalized such that the average IgG1$^a$/IgG1$^b$ ratio in Balb/c-B6 F1 samples = 1.0. For (A-D), statistics from n = 4 unimmunized mice of each genotype and n = 3 WT, n = 6 $IgM^{-/-}$, and n = 7 $IgD^{-/-}$ immunized mice were pooled. For (E-F), n = 5 unimmunized and n = 5 immunized mice are shown. For (G-I), n = 5 Balb/c-B6 F1 mice and n = 3 $IgM^{+/-}$ mice are shown. One-way ANOVA with Tukey's multiple comparisons test (B and D), a paired t test (E-F), and Welch's t test (G-I) were used to calculate p values, and mean +SEM is displayed. *p<0.05, **p<0.01, ***p<0.001.

DOI: https://doi.org/10.7554/eLife.35074.035

The following source data and figure supplements are available for figure 7:

**Source data 1.** Numerical data corresponding to germinal center and plasma cell responses in *Figure 7B-I*,*Figure 7—figure supplement 1A–D*
DOI: https://doi.org/10.7554/eLife.35074.038

**Figure supplement 1.** Role of BCR allotype and generation of unswitched plasma cells in SRBC-immunized mice.
DOI: https://doi.org/10.7554/eLife.35074.036

**Figure supplement 2.** Role of IgM and IgD in regulating rapid antibody responses.
DOI: https://doi.org/10.7554/eLife.35074.037

Fc portions of IgM (*Übelhart et al., 2015*). This could account for selective quiescence of mildly autoreactive B cells to monovalent endogenous antigens. However, the identity and structural characteristics of endogenous antigens are not well understood, and include cell-surface antigens on the surface of red cells and B cells (*Quách et al., 2011*; *Reed et al., 2016*). Moreover, recent work from Goodnow and colleagues showed that HEL-specific IgD BCR, when expressed on primary mouse splenocytes, can indeed mobilize calcium in vitro and can mediate gene expression changes in vivo in response to monovalent cognate antigen (*Sabouri et al., 2016*).

How do our observations help move beyond and reconcile the seemingly contradictory published work on antigen sensing by IgM and IgD? In vitro signaling studies of HEL-specific BCRs by the Jumaa and Goodnow labs differ in several important ways; the former work assesses calcium signaling in pre-B cells that lack Slp-65 expression, while the latter studies splenic B cells with a considerably more mature phenotype. In addition, differences in reagents or strength of stimulus may also

contribute to opposing conclusions. Our data suggests that the reduced sensitivity of IgD to bona fide endogenous antigen may lie somewhere between these two extremes; we find that IgD can sense endogenous antigens, just less efficiently than IgM. Consistent with this conclusion, both HEL-specific IgM and IgD BCRs can mediate B cell anergy and drive a common gene expression program in response to chronic high affinity soluble antigen exposure in vivo (*Brink et al., 1992*; *Sabouri et al., 2016*).

Here we show that B cells expressing IgD BCR alone inefficiently sense endogenous antigens, and exhibit impaired in vivo cell fate decisions that are dependent upon BCR signal strength. This is epitomized by dramatic skewing of IgD-only B cells away from the B1a and towards the MZ fate in a competitive setting. Importantly, signal transduction mediated by the IgD BCR in response to anti-Igκ stimulation in vitro is intact for all assayed parameters, suggesting that antigen sensing and signal initiation rather than signal transduction downstream of IgD is impaired. We propose that this impaired sensing is conferred by structural properties of IgD, either through direct binding of antigen and signal transduction through the BCR or through differential pairing with co-receptors that directly modulate the BCR signaling pathway.

Characterization of bona fide endogenous antigens that drive Nur77-eGFP expression in the polyclonal repertoire is necessary in order to further dissect the biophysical basis for reduced antigen sensing by IgD in vivo. These antigens may not be exclusively restricted to those that are germline-encoded or generated by host enzymes; rather they may include non-inflammatory antigens taken up through the gastrointestinal tract and recognized by specific BCRs. At least some relevant endogenous antigens may be membrane-bound (*Quách et al., 2011*; *Reed et al., 2016*), and recent studies have illustrated the importance of membrane spreading, contraction, and stiffness discrimination in BCR signaling (*Fleire et al., 2006*; *Shaheen et al., 2017*). We propose that relevant endogenous antigens are presented in contexts that do not efficiently trigger signaling through the flexible structure of IgD. This might explain the discrepancy between weak responsiveness of IgD to endogenous antigens in vivo and strong signaling in response to crosslinking antibodies in vitro.

An alternative mechanism is that differential clustering of co-receptors with IgD and IgM might influence how BCR signals are integrated in vivo. It has long been proposed that there are not only quantitative, but also qualitative differences between IgM and IgD signaling. Early studies of IgM and IgD signaling in cell lines suggested that kinetics but not quality of signaling triggered by each isotype differed (*Kim and Reth, 1995*). However, we have been unable to identify a difference in kinetics of BCR signaling in primary IgM- or IgD-deficient B cells in vitro. More recently, studies using TIRF, dSTORM, and PLA (proximity ligation assay) showed that IgD is more densely clustered on the cell surface than IgM, and these distinct IgM- and IgD-containing 'islands' are differentially associated with co-receptors such as CD19 in resting and activated B cells (*Kläsener et al., 2014*; *Maity et al., 2015*; *Mattila et al., 2013*). Since function of the ITIM-containing inhibitory co-receptor CD22 depends on its ability to efficiently access the BCR, differences in cluster density of IgM and IgD might render them differentially sensitive to such inhibitory tone (*Gasparrini et al., 2016*). It is possible that differential association with such co-receptors contributes to differences in the function of the IgM and IgD BCRs in vivo by modulating BCR signal strength, or by selectively perturbing specific downstream signaling events.

In addition to B cell co-receptors that have long been known to directly modulate canonical BCR signaling, a growing list of immunoreceptors expressed on B cells have more recently been shown to require expression of the BCR for optimal function; Becker et al. have shown that expression of the IgD BCR is critical for CXCR4-dependent signaling (*Becker et al., 2017*). This may influence the biology of B cells expressing only IgM; indeed, CXCR4-deficient B cells exhibit reduced plasma cell migration from spleen to bone marrow (but intact splenic SLPC responses) (*Nie et al., 2004*). Importantly, defective CXCR4 signaling in IgM-only B cells could not account for reduced Nur77-eGFP expression in IgD-only B cells because reporter expression is insensitive to CXCR4 signaling (*Figure 1—figure supplement 1C*). Nor would it account for skewed cell fate decisions by IgD-only B cells in competition with wild type B cells, both of which retain intact CXCR4 signaling. B cell responses to TLR4 ligands also require expression of the BCR, but importantly, canonical TLR ligands do not exhibit selective dependence on either the IgM or the IgD isotypes (*Figure 1—figure supplement 3A*). Finally, recent work establishes a role for the Fc-receptor for IgM in modulating B cell responses, and could therefore play a role downstream of serum IgM that is absent in $IgM^{-/-}$ mice

(*Nguyen et al., 2017a*; *Nguyen et al., 2017b*). We control for this effect by confirming that all phenotypes in this study are cell-intrinsic and independent of secreted IgM.

Accumulating evidence suggests that strong BCR signals favor SLPC over GC fate (*Chan et al., 2009*; *Nutt et al., 2015*; *Paus et al., 2006*). This is thought to be transcriptionally mediated at least in part by loss of Ets1 expression and induction of Irf4 in a BCR signal-strength dependent manner (*Nutt et al., 2015*). Here we show that IgM and IgD are each sufficient to drive GC responses, but IgD is less efficient at promoting SLPC fate in response to endogenous antigens, implying that IgD$^{hi}$ IgM$^{lo}$ B cells may similarly be shunted away from SLPC responses. This discrepancy is unlikely to be attributable to differences in antigen capture and presentation by the IgM and IgD BCRs, as GC entry is highly T cell-help dependent. Instead, we provide evidence to suggest that reduced antigen-dependent BCR signals are transduced in vivo by IgD. We show that, in the absence of Lyn, Ets1 downregulation and unswitched PC expansion require the IgM BCR. $IgM^{-/-}Lyn^{-/-}$ mice phenocopy $Btk^{lo}Lyn^{-/-}$ mice; both strains are protected from Ets1-downregulation, PC expansion, and anti-dsDNA autoantibody production in vivo (*Mayeux et al., 2015*; *Whyburn et al., 2003*). This suggests that robust Btk-dependent Ets-1 downregulation in response to endogenous antigens relies upon efficient antigen sensing by the IgM BCR and is important to drive unswitched PC expansion in the absence of Lyn. It will be important to explore whether a similar mechanism accounts for impaired IgG1$^+$ SLPC responses by Lyn-sufficient B cells with low or absent IgM expression.

We propose that the purpose (and consequence) of IgM downregulation on autoreactive B cells is to limit their direct differentiation into SLPCs in response to chronic endogenous antigen stimulation, and prevent secretion of auto-antibodies in the context of an acute humoral immune response (*Figure 7—figure supplement 2A*). However, IgD is sufficient to drive germinal center differentiation without the contribution of IgM. This has been well-demonstrated both in the present study as well as prior work (*Lutz et al., 1998*). Indeed, not only are autoreactive IgD$^{hi}$ IgM$^{lo}$ B cells competent to enter the germinal center, they appear to do so with greater efficiency, perhaps due in part to improved survival (*Sabouri et al., 2016*; *Sabouri et al., 2014*). It is worth emphasizing that selection pressures and tolerance mechanisms operating within the germinal center must independently ensure that somatic hypermutation both abolishes germ-line-encoded autoreactivity and prevents de novo acquisition of autoreactivity. Goodnow and colleagues recently showed that autoreactive BCRs can indeed be 'redeemed' by somatic hypermutation, providing proof-of-principle that entry of autoreactive B cells into the GC does not necessarily pose a risk to the organism (*Sabouri et al., 2014*).

Why, though, are autoreactive B cells preserved in the periphery and not deleted? Why is IgD necessary at all? One suggestion is that such BCRs are retained in the pre-immune B cell compartment to fill holes in the repertoire, and that IgD is essential for their survival; indeed, it has been shown that loss of IgD expression, with or without compensatory upregulation of IgM, leads to loss of mature naïve B cells in competition (*Roes and Rajewsky, 1993*; *Sabouri et al., 2016*). We similarly find that IgD-only B cells have a profound competitive advantage in the follicular B cell compartment relative to IgM-only (but not WT) B cells, implying an obligate survival function for the IgD isotype BCR. This is consistent with a well-appreciated pro-survival function of the BCR as demonstrated by Rajewsky and colleagues (*Kraus et al., 2004*; *Lam et al., 1997*). Since IgD expression is needed to keep IgM$^{lo}$ B cells alive, this may explain why IgD expression facilitates a broad dynamic range of IgM expression across the B cell repertoire (*Figure 1B,C*), allowing B cells to fine-tune the intensity of their SLPC responses according to their autoreactivity. IgD expression on IgM$^{lo}$ cells could serve to mediate antigen capture and participation in T-dependent immune responses, while simultaneously limiting SLPC responses as we show here.

We demonstrate here for the first time that IgD BCRs sense endogenous antigen more weakly than IgM BCRs in vivo, are inefficient at driving SLPC responses to endogenous antigens, and thereby may function to divert autoreactive follicular B cells from direct PC differentiation. We propose that this property of the IgD BCR limits generation of auto-reactive antibodies in the context of immediate humoral immune responses. Taken together with recent work from the Jumaa and Goodnow groups, we propose a unified model of how IgM downregulation in the face of high IgD expression regulates the fate of naïve autoreactive B cells in vivo while retaining their contribution to the mature BCR repertoire.

# Materials and methods

## Key resources table

| Reagent type (species) or resource | Designation | Source or reference | Identifiers | Additional information |
|---|---|---|---|---|
| Strain, strain background (*Mus musculus*) | C56BL/6 | The Jackson Laboratory; Taconic | JAX:000664; TAC:BCNTac | |
| Strain, strain background (*M. musculus*) | Balb/C | The Jackson Laboratory | JAX:000651 | |
| Strain, strain background (*M. musculus*) | Nur77-eGFP | MMRRC UC Davis | MMRRC:012015-UCD | Characterized in PMID:22902503 |
| Strain, strain background (*M. musculus*) | IgHEL | PMID:3261841 | | MD-4 |
| Strain, strain background (*M. musculus*) | sHEL | PMID:3261841 | | ML-5 |
| Strain, strain background (*M. musculus*) | IgM-/- | PMID:9655395 | | |
| Strain, strain background (*M. musculus*) | IgD-/- | PMID:8446604 | | |
| Strain, strain background (*M. musculus*) | CD40L-/- | PMID:7964465 | | |
| Strain, strain background (*M. musculus*) | Unc93b 3d/3d | PMID:16415873 | | |
| Strain, strain background (*M. musculus*) | TLR7-/- | PMID:15034168 | | |
| Strain, strain background (*M. musculus*) | BaffTg | MMRRC UC Davis | RRID:MMRRC_036508-UCD | Described in PMID:15972664 |
| Strain, strain background (*M. musculus*) | B6.IgHa | The Jackson Laboratory | JAX:001317 | B6.Cg-Gpi1a Thy1a Igha/J |
| Strain, strain background (*M. musculus*) | Lyn-/- | PMID:9252121 | | |
| Strain, strain background (*M. musculus*) | MB1-Cre | PMID:16940357 | | |
| Strain, strain background (*M. musculus*) | MyD88 fl/fl | PMCID:PMC2847796 | | |
| Strain, strain background (*M. musculus*) | Nr4a1-/- | PMID:7624775 | JAX:006187 | |
| Biological sample (*Ovis aires*) | Sheep Red Blood Cells | Rockland | R406-0050 | |
| Antibody | Anti-B220-A647 (rat monoclonal) | BD Pharmingen | 557683 | (1:200) |
| Antibody | Anti-B220-APC-e780 (rat monoclonal) | eBioscience | 47-0452-82 | (1:400) |
| Antibody | Anti-B220-FITC (rat monoclonal) | Tonbo | 35–0452 U100 | (1:200) |
| Antibody | Anti-B220-Pacific Blue (rat monoclonal) | Tonbo | 75–0452 U100 | (1:200) |
| Antibody | Anti-B220-PE (rat monoclonal) | BD Pharmingen | 553090 | (1:200) |
| Antibody | Anti-B220-PE-Cy7 (rat monoclonal) | BD Pharmingen | 552772 | (1:200) |
| Antibody | Anti-B220-PerCP-Cy5.5 (rat monoclonal) | Tonbo | 65–0452 U100 | (1:200) |
| Antibody | Anti-CD5-APC (rat monoclonal) | Tonbo | 20–0051 U100 | (1:100) |

*Continued on next page*

*Continued*

| Reagent type (species) or resource | Designation | Source or reference | Identifiers | Additional information |
|---|---|---|---|---|
| Antibody | Anti-CD19-PE-Cy7 (rat monoclonal) | Biolegend | 115520 | (1:150) |
| Antibody | Anti-CD19-PerCP-Cy5.5 (rat monoclonal) | BD Pharmingen | 551001 | (1:150) |
| Antibody | Anti-CD21-A647 (rat monoclonal) | Biolegend | 123424 | (1:100) |
| Antibody | Anti-CD21-Pacific Blue (rat monoclonal) | Biolegend | 123414 | (1:100) |
| Antibody | Anti-CD23-A647 (rat monoclonal) | Biolegend | 101612 | (1:200) |
| Antibody | Anti-CD23-FITC (rat monoclonal) | BD Pharmingen | 553138 | (1:200) |
| Antibody | Anti-CD23-Pacific Blue (rat monoclonal) | Biolegend | 101616 | (1:100) |
| Antibody | Anti-CD23-PE (rat monoclonal) | BD Pharmingen | 553139 | (1:200) |
| Antibody | Anti-CD23-PE-Cy7 (rat monoclonal) | eBioscience | 25-0232-82 | (1:200) |
| Antibody | Anti-CD69-APC (hamster monoclonal) | Biolegend | 104514 | (1:100) |
| Antibody | Anti-CD69-PE-Cy7 (hamster monoclonal) | Tonbo | 60–0691 U100 | (1:100) |
| Antibody | Anti-CD86-Pacific Blue (rat monoclonal) | Biolegend | 105022 | (1:100) |
| Antibody | Anti-CD93 (AA4.1)-PE-Cy7 (rat monoclonal) | Biolegend | 136506 | (1:100 |
| Antibody | Anti-CD138-PE (rat monoclonal) | Biolegend | 142504 | (1:100) |
| Antibody | Anti-CD138-PE-Cy7 (rat monoclonal) | Biolegend | 142513 | (1:100) |
| Antibody | Anti-CXCR4-Biotin (rat monoclonal) | BD Pharmingen | 551968 | (1:100) |
| Antibody | Anti-ETS1 (rabbit monoclonal) | Epitomics; abcam | EPI:3123–1; AB:109212 | (1:10,000); concentrated lot from L.A. Garrett-Sinha |
| Antibody | Anti-Fas-PE-Cy7 (hamster monoclonal) | BD Pharmingen | 557653 | (1:200) |
| Antibody | Anti-GAPDH (mouse monoclonal) | EMD Millipore | AB2302 | (1:2000) |
| Antibody | GL-7-A647 (rat monoclonal) | BD Biosciences | 561529 | (1:400) |
| Antibody | Anti-IgA-Biotin (rat monoclonal) | Biolegend | 407003 | (1:400) |
| Antibody | Anti-IgD (goat polyclonal serum) | MD Biosciences | 2057001 | (1:50-1:400) |
| Antibody | Anti-IgD[a]-Biotin (mouse monoclonal) | BD Pharmingen | 553506 | (1:300) |
| Antibody | Anti-IgD-APC-e780 (rat monoclonal) | eBioscience | 47-5993-80 | (1:500) |
| Antibody | Anti-IgD-HRP (rat monoclonal) | American Research Products | 09-1008-4 | (1:2000) |
| Antibody | Anti-IgD-Pacific Blue (rat monoclonal) | Biolegend | 405712 | (1:300) |

*Continued*

| Reagent type (species) or resource | Designation | Source or reference | Identifiers | Additional information |
|---|---|---|---|---|
| Antibody | Anti-IgD-PE (rat monoclonal) | eBioscience | 12-5993-82 | (1:800) |
| Antibody | Anti-IgG1[a]-Biotin (mouse monoclonal) | BD Pharmingen | 553500 | (1:300) |
| Antibody | Anti-IgG1[b]-Biotin (mouse monoclonal) | BD Pharmingen | 553533 | (1:300) |
| Antibody | Anti-IgG2a[a]-Biotin (mouse monoclonal) | BD Biosciences | 553502 | (1:1000) |
| Antibody | Anti-IgG2a[b]-Biotin (mouse monoclonal) | BD Biosciences | 553504 | (1:1000) |
| Antibody | Anti-IgG2c-Biotin (goat polyclonal) | SouthernBiotech | 1079–08 | (1:1000) |
| Antibody | Anti-Igκ (goat polyclonal) | SouthernBiotech | 1050–01 | 1–20 µg/mL |
| Antibody | Anti-Igκ-F(ab')2 (goat polyclonal) | SouthernBiotech | 1052–01 | 1–20 µg/mL |
| Antibody | Anti-Igκ-FITC (goat polyclonal) | SouthernBiotech | 1050–02 | (1:300) |
| Antibody | Anti-Igκ-FITC (rat monoclonal) | BD Pharmingen | 550003 | (1:300) |
| Antibody | Anti-Igk-PerCP-Cy5.5 (rat monoclonal) | BD Pharmingen | 560668 | (1:300) |
| Antibody | Anti-Igλ-FITC (rat monoclonal) | BD Pharmingen | 553434 | (1:300) |
| Antibody | Anti-Igλ-PE (rat monoclonal) | Biolegend | 407307 | (1:300) |
| Antibody | Anti-IgM-F(ab')2 (goat polyclonal) | Jackson ImmunoResearch | 115-006-020 | 1–20 µg/mL |
| Antibody | Anti-IgM-HRP (goat polyclonal) | SouthernBiotech | 1020–05 | (1:2000); secondary for ELISA |
| Antibody | Anti-IgM[a]-Biotin (mouse monoclonal) | Biolegend | 408603 | (1:100) |
| Antibody | Anti-IgM[a]-FITC (mouse monoclonal) | Biolegend | 408606 | (1:100); (1:400) for plasma cell |
| Antibody | Anti-IgM[a]-PE (mouse monoclonal) | BD Pharmingen | 553517 | (1:100); (1:400) for plasma cell |
| Antibody | Anti-IgM[b]-Biotin (mouse monoclonal) | Biolegend | 406204 | (1:100) |
| Antibody | Anti-IgM[b]-PE (mouse monoclonal) | Biolegend | 406208 | (1:100); (1:400) for plasma cell |
| Antibody | Anti-IgM-APC (rat monoclonal) | eBioscience | 17-5790-82 | (1:100) |
| Antibody | Anti-MHC-2-APC (rat monoclonal) | Tonbo | 20–5321 U100 | (1:1000) |
| Antibody | Anti-Mouse-IgG(H + L)-HRP (goat polyclonal) | SouthernBiotech | 1031–05 | (1:5000); secondary for western blots |
| Antibody | Anti-Nur77-PE (mouse monoclonal) | eBioscience | 12-5965-80 | (1:100) |
| Antibody | Anti-pERK (rabbit monoclonal) | Cell Signaling Technology | 4377S | (1:80) |
| Antibody | Anti-pS6 (rabbit monoclonal) | Cell Signaling Technology | 4856S | (1:100) |
| Antibody | Anti-Rabbit-IgG-APC (donkey polyclonal) | Jackson ImmunoResearch | 711-136-152 | (1:100); secondary for pERK/pS6 |

*Continued*

| Reagent type (species) or resource | Designation | Source or reference | Identifiers | Additional information |
|---|---|---|---|---|
| Antibody | Anti-Rabbit-IgG(H + L)-HRP (goat polyclonal) | SouthernBiotech | 4050–05 | (1:5000); secondary for western blots |
| Sequence-based reagent | Nr4a1 forward primer | Elim Biopharm | | gcctagcactgccaaattg |
| Sequence-based reagent | Nr4a1 reverse primer | Elim Biopharm | | ggaaccagagagcaagtcat |
| Sequence-based reagent | GAPDH forward primer | Elim Biopharm | | aggtcggtgtgaacggatttg |
| Sequence-based reagent | GAPDH reverse primer | Elim Biopharm | | tgtagaccatgtagttggaggtca |
| Peptide, recombinant protein | NP-RSA | Biosearch | N-5054–100 | Conj. ratio: 10 |
| Peptide, recombinant protein | NP-BSA | Biosearch | N-5050H-100 | Conj. ratio: 23 |
| Peptide, recombinant protein | Streptavidin-HRP | SouthernBiotech | 7100–05 | (1:5000) |
| Peptide, recombinant protein | Streptavidin-APC | Tonbo | 20–4317 U500 | (1:100-1:400) |
| Peptide, recombinant protein | Streptavidin-Pacific Blue | Life Technologies | S11222 | (1:200) |
| Peptide, recombinant protein | Streptavidin-PerCP-Cy5.5 | BD Pharmingen | 551419 | (1:400) |
| Peptide, recombinant protein | Streptavidin-FITC | Biolegend | 405202 | (1:100-1:200) |
| Peptide, recombinant protein | CXCL12 | Peprotech | 300-28A | |
| Peptide, recombinant protein | NP-PE | Biosearch | N-5070–1 | (1:400) |
| Chemical compound, drug | Poly-L-Lysine | Sigma | P2636-100MG | 100 µg/mL in 0.1 M Tris-HCl pH7.3 |
| Chemical compound, drug | Poly dA-dT | Sigma | P0883-50UN | 0.2 U/mL in 0.1 M Tris-HCl pH7.3 |
| Chemical compound, drug | LPS | Sigma | L8274 | |
| Chemical compound, drug | CpG | InvivoGen | tlrl-1826b | |
| Chemical compound, drug | Pam3CSK4 | InvivoGen | tlrl-pms | |
| Commercial assay, kit | Indo-1, AM | Life Technologies | I-1223 | (1:1000) |
| Commercial assay, kit | Live/Dead Fixable Near-IR Dead Cell Stain Kit | Invitrogen | L10119 | (1:1000) |
| Commercial assay, kit | ECL Luminol; Oxidizer Reagents | Perkin Elmer | 0RT2751; 0RT2651 | |
| Commercial assay, kit | 3,3',5,5'-Tetramethylbenzidine, Slow Kinetic Form | Sigma | T4319-100ML | ELISA substrate |
| Software, algorithm | FlowJo | FlowJo LLC | | Version 9.9.4 |
| Software, algorithm | Prism | GraphPad | | Version 7.0b |
| Software, algorithm | Canopy | Enthought | | Version 1.4.1.1975 |
| Software, algorithm | Binning program in *Figure 1F* | Other | | Source code provided in this publication |
| Other | BD Microtainer Capillary Blood Collector | Fisher | 365967 | |
| Other | PtC-Rhodamine (DOPC/CHOL Liposomes) | FormuMax | F60103F-R | (1:1000); used in PtC-specific B1a staining |
| Other | NuPAGE 4–12% Bis-Tris Protein Gels | Invitrogen | NP0335BOX | |
| Other | Immobilon-P PVDF Membrane | EMD Millipore | IPVH00010 | |
| Other | Assay Plate, 96 Well, No Lid, Vinyl | Costar | 2595 | Used for ELISA |

## Mice

Nur77-eGFP mice and $Lyn^{-/-}$ mice have been previously described (*Chan et al., 1997*; *Zikherman et al., 2012*). BAFF Tg mice were originally generated in the Nemazee lab, and express BAFF under the control of the myeloid promoter human CD68 (*Gavin et al., 2005*) (source: MMRRC UC Davis). $IgD^{-/-}$ and $IgM^{-/-}$ mice were previously described, and the former were generously shared by Dr. Hassan Jumaa (*Lutz et al., 1998*; *Nitschke et al., 1993*). $TLR7^{-/-}$, $CD40L^{-/-}$, $Unc93b1^{3d/3d}$, $Nr4a1^{-/-}$, and MB1-Cre $MyD88^{fl/fl}$ mice were previously described (*Hobeika et al., 2006*; *Hou et al., 2008*; *Lee et al., 1995*; *Lund et al., 2004*; *Renshaw et al., 1994*; *Tabeta et al., 2006*). All strains were backcrossed to the C57BL/6 genetic background for at least six generations. Mice were used at 5–12 weeks of age for all functional and biochemical experiments unless otherwise noted. Germ-free (GF) and specific pathogen free (SPF) C57BL/6 mice used for direct comparison were purchased from Taconic and Jackson Laboratory and kept in microisolators under GF or SPF (ISOcage P - Bioexclusion System, Tecniplast) conditions. All GF mice were housed in closed caging systems and provided with standard irradiated chow diet, acidified water and housed under a 12 hr light cycle; 7-week-old males mice were used. GF and control mice were generously provided by Drs. Sergio Baranzini and Anne-Katrin Proebstel at UCSF. All other mice used in our studies were housed in a specific pathogen free facility at UCSF according to the University Animal Care Committee and National Institutes of Health (NIH) guidelines.

## Antibodies and reagents

Fluorescently-conjugated or biotin-conjugated antibodies to B220, CD5, CD19, CD21, CD23, CD69, CD86, CD93 (AA4.1), CD95 (Fas), CD138, CXCR4, IgA, IgM, IgM[a], IgM[b], IgD, IgD[a], IgG1[a], IgG1[b], Igκ, Igλ, GL-7, MHC Class II, and fluorescently-conjugated streptavidin were from Biolegend, eBiosciences, BD Biosciences, Tonbo, or Life Technologies. NP-PE was from Biosearch Technologies. 100 nm Rhodamine PtC liposomes were from FormuMax. Antibodies for intra-cellular staining, pErk Ab (clone 194g2) and pS6 Ab (2F9), were from Cell Signaling Technologies, and Nur77 Ab (clone 12.14) conjugated to PE was from eBioscience. Goat anti-mouse IgM F(ab')$_2$ was from Jackson Immunoresearch. Goat anti-mouse Igκ and goat F(ab')$_2$ anti-mouse Igκ were from Southern Biotech. Anti-IgD was from MD Biosciences. CXCR4 ligand (CXCL12/hSDF-1α) was from Peprotech. LPS (Cat. L8274) was from Sigma. CpG DNA (ODN 1826 Biotin; Cat. tlrl-1826b) and Pam3CSK4 (Cat. tlrl-pms) were from InvivoGen.

## Flow cytometry and data analysis

Cells were stained with indicated antibodies and analyzed on a Fortessa (Becton Dickson) as previously described (*Hermiston et al., 2005*). Data analysis was performed using FlowJo (v9.7.6) software (Treestar Incorporated, Ashland, OR). Statistical analysis and graphs were generated using Prism v6 (GraphPad Software, Inc). Figures were prepared using Illustrator CS6 v16.0.0. Median Igk levels (*Figure 1F*) were calculated for 200 equally sized 'bins' spanning the Nur77-eGFP spectrum using Canopy v1.4.1 (Enthought); source code is provided, and parameters can be modified to condense any 2D FACS plot for comparisons of multiple samples.

## Statistics and replicates

We define biological replicates as independent analyses of cells isolated from different mice of the same genotype, and we define technical replicates as analyses of cells isolated from the same mouse and used in the same experiment. When technical replicates were used, we averaged them to calculate a value for the biological replicate. All reported values and statistics correspond to biological replicates only, and all 'n' values reported reflect the number of biological replicates. Wherever MFIs are directly compared, we collected all samples in a single experiment to avoid potential error that could arise from fluctuations in our flow cytometers. In some instances, MFI ratios were compiled from different experiments for statistical purposes, but qualitative findings were consistent experiment to experiment. As our panels reproducibly generated flow plots with well-defined populations, population sizes were calculated from data compiled from different experiments. When comparing two groups, we employed Welch's t test, which is more stringent than Student's t test and does not assume that the groups have the same standard deviation. We used paired difference t tests when studying parameters in allotype-heterozygous mice that are sensitive to cell-extrinsic factors (e.g.

dose of immunogen or severity of autoimmune disease). When comparing three genotypes (e.g. receptor levels on WT vs. $IgM^{-/-}$ vs. $IgD^{-/-}$ B cells), we used one-way ANOVA and Tukey's multiple comparison test to calculate p values.

## Intracellular Nur77, pErk and pS6 staining

Ex vivo or following stimulation, cells were fixed in 2% paraformaldehyde, permeabilized with 100% methanol, and stained with anti-Nur77, anti-pErk, or anti-pS6 followed by lineage markers and secondary antibodies if needed.

## Intracellular plasma cell staining

Cells were fixed in 2% paraformaldehyde and permeabilized in BD Perm/Wash. Stains for antibody isotypes and allotypes were prepared in BD Perm/Wash.

## Intracellular calcium flux

Cells were loaded with 5 µg/mL Indo-1 AM (Life Technologies) and stained with lineage markers. Cells were rested at 37 C for 2 min, and Indo-1 fluorescence was measured immediately prior to stimulation to calculate basal calcium, and for at least two minutes following addition of stimulus.

## In vitro B cell stimulation

Splenocytes or lymphocytes were harvested into single cell suspension, subjected to red cell lysis using ACK buffer, and plated at a concentration of $7.5 \times 10^5$ cells/200 µL in round bottom 96 well plates in complete RPMI media with stimuli for 18 hours prior to analysis. In vitro cultured cells were stained using fixable near IR live/dead stain (Life technologies) per manufacturer's instructions.

## ELISA

Serum antibody titers for total IgM, total IgD, anti-dsDNA IgG2a, and NP-specific IgG1 were measured by ELISA. For total IgM and total IgD, 96-well plates (Costar) were coated with 1 µg/mL anti-IgM F(ab')$_2$ (Jackson) or 1 µg/mL anti-IgD (BD 553438), respectively. Sera were diluted serially, and total IgM and total IgD were detected with anti-IgM-biotin (eBioscience) and SA-HRP (Southern Biotech) or anti-IgD-HRP (American Research Products). dsDNA plates were generated by serially coating plates with 100 µg/mL poly-L-lysine (SigmaAldrich) and 0.2 U/mL poly dA-dT (SigmaAldrich) in 0.1 M Tris-HCL pH 7.6. Sera were diluted serially on dsDNA plates, and autoantibodies were detected with IgG2[a]-biotin, IgG2a[b]-biotin, and SA-HRP. NP plates were generated by coating 96-well plates with 1 µg/mL NP-BSA (Biosearch, conjugation ratio 23) in PBS. Sera from NP-RSA immunized and unimmunized mice were diluted serially, and NP-specific IgG1 was detected with IgG1[a]-biotin, IgG1[b]-biotin, and SA-HRP. All ELISA plates were developed with TMB (Sigma) and stopped with 2N sulfuric acid. Absorbance was measured at 450 nm. For total IgM, total IgD, and dsDNA IgG2a, OD values or OD ratios were calculated and displayed. For NP-specific IgG1, titers were calculated at OD = 0.2 and normalized such that the average IgG1[a]/IgG1[b] ratio in immunized Balb/c-C57BL/6 F1 mice equals 1.0.

## B cell purification and western blots

Splenic B cells were purified by negative selection with a MACS kit according to manufacturer's protocol (Miltenyi Biotech, cat# 130-090-862). Purified B cells were lysed in 2N 4X SDS and boiled at 95 C for 5 min. 500,000 cells per lane were loaded onto NuPAGE 4–12% Bis-Tris gels (Invitrogen NP0335), run for 50 min at 200V, and transferred onto PVDF membranes using an XCell II Blot Module (Invitrogen). Membranes were blocked with 3% BSA. Proteins were detected with rabbit monoclonal anti-Ets1 (Epitomics, custom lot provided by Lee Ann Garrett-Sinha), mouse anti-mouse GAPDH (Santa Cruz Biotechnology 32233), and HRP-conjugated secondary antibodies goat anti-rabbit IgG (H + L) and goat anti-mouse IgG (H + L) (SouthernBiotech). Membranes were developed with Western Lightning Plus ECL (Perkin Elmer 0RT2651 and 0RT2751) and imaged using a ChemiDoc Touch Imaging System (Bio-Rad). Quantifications were performed using Image Lab v. 5.2.14 (Bio-Rad).

## Immunizations

Mice were immunized i.p. with 200 uL of 10% sheep red blood cells (Rockland R406-0050) diluted in PBS. Mice were sacrificed 5 days after immunization, and serum and spleens were harvested. Mice were immunized i.p. with 100 ug of NP-conjugated rabbit serum album (NP-RSA, Biosearch N-5054–10, conjugation ratio 10) prepared in PBS with Alhydrogel 1% adjuvant (Accurate Chemical and Scientific Corp. 21645-51-2). NP-RSA immunized mice were sacrificed 7–8 days following immunization, and serum and spleens were harvested. To ensure that our calculated values accurately reflect the magnitude and variability of immune responses induced by immunization, we analyzed all mice that had larger plasma cell and germinal center compartments than unimmunized mice. One $IgM^{+/-}$ mouse immunized with SRBCs (*Figure 7E–F*) was excluded because it did not display an expanded plasma cell compartment; no other mice were excluded from analysis.

## Quantitative PCR

Splenocytes were lysed in TRIzol (Invitrogen) and stored at $-80$ C. cDNA was prepared using a SuperScriptIII kit (Invitrogen). Quantitative PCR reactions were run on a QuantStudio 12K Flex thermal cycler (Applied Biosystems) with FastStart Universal SYBR Green Master Mix (Roche). Nr4a1 (forward: gcctagcactgccaaattg; reverse: ggaaccagagagcaagtcat) and GAPDH (forward: aggtcggtgtgaacggatttg; reverse: tgtagaccatgtagttgaggtca) primers were used at 250 nM each.

## Acknowledgements

We thank Anthony DeFranco, Arthur Weiss, and Jason Cyster for valuable feedback and suggestions for this manuscript, and Al Roque for help with mouse husbandry. We thank Renuka Nayak, Anne-Katrin Proebstel, and Eric Wigton for sharing valuable mouse reagents. Funding for this project was provided by the Rheumatology Research Foundation (JZ), NIAMS K08 AR059723 and R01 A127648 (JZ), and NSF Graduate Research Fellowship 1650113 (MN).

## Additional information

### Funding

| Funder | Grant reference number | Author |
| --- | --- | --- |
| National Science Foundation | 1650113 | Mark Noviski |
| National Institute of Arthritis and Musculoskeletal and Skin Diseases | R01 A127648 | Julie Zikherman |
| Rheumatology Research Foundation | | Julie Zikherman |
| National Institute of Arthritis and Musculoskeletal and Skin Diseases | NIAMS K08 AR059723 | Julie Zikherman |

The funders had no role in study design, data collection and interpretation, or the decision to submit the work for publication.

### Author contributions

Mark Noviski, Conceptualization, Data curation, Formal analysis, Funding acquisition, Investigation, Methodology, Writing—original draft, Writing—review and editing; James L Mueller, Formal analysis, Investigation; Anne Satterthwaite, Lee Ann Garrett-Sinha, Frank Brombacher, Resources, Writing—review and editing; Julie Zikherman, Conceptualization, Data curation, Formal analysis, Supervision, Funding acquisition, Investigation, Methodology, Writing—original draft, Project administration, Writing—review and editing

## Author ORCIDs

Mark Noviski (iD) https://orcid.org/0000-0001-8072-1059
Julie Zikherman (iD) http://orcid.org/0000-0002-0873-192X

## Ethics

Animal experimentation: This study was performed in strict accordance with the recommendations in the Guide for the Care and Use of Laboratory Animals of the National Institutes of Health. All of the animals were handled according to approved institutional animal care and use committee (IACUC) protocols of the University of California, San Francisco. The protocol was approved by the IACUC committee of the University of California, San Francisco (Protocol number: AN171020-01).

## Decision letter and Author response

Decision letter https://doi.org/10.7554/eLife.35074.051
Author response https://doi.org/10.7554/eLife.35074.052

## Additional files

### Supplementary files

• Transparent reporting form
DOI: https://doi.org/10.7554/eLife.35074.039

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
