## [Decision Letter]

[Editors’ note: a previous version of this study was rejected after peer review, but the authors submitted for reconsideration. The first decision letter after peer review is shown below.]

Thank you for submitting your work entitled "Differential sensing of endogenous antigens and control of B cell fate by IgM and IgD B cell receptors" for consideration by *eLife*. Your article has been reviewed by two peer reviewers, and the evaluation has been overseen by a Reviewing Editor and a Senior Editor. The reviewers have opted to remain anonymous.

Our decision has been reached after consultation between the reviewers. As we feel the additional work needed to address the issues raised by the reviewers would take more than two months to complete, we are returning your submission to you now in case you wish to submit elsewhere for speedy publication. However, if you address these points and wish to resubmit your work to *eLife*, we would be happy to look at a revised paper. Please note that it would be treated as a new submission with no guarantees of acceptance.

Reviewer #2:

The manuscript entitled "Differential sensing of endogenous antigens and control of B cell fate by IgM and IgD B cell receptors" by Noviski et al. follows a series of manuscripts started back in 2012 with the publication of the Nur77-GFP mice (Zikherman et al., 2012). Nur77-GFP mice have thus emerged as a useful tool to study B cell antigen receptor (BCR) and T cell antigen receptor (TCR) signaling in vivo.

In the present manuscript, the authors aimed to answer the long-standing question why mature B cells express two BCR isotypes, IgM and IgD, with identical antigen-specificity, and how these two isotypes impose B cell tolerance in vivo.

A major issue for me is the clarity of the present manuscript. Although the initial question is clear and well defined, a good rational flow along the experiments, figures and the text is missing. Also the way the data are presented is highly confusing and prevents comparison between figures. The manuscript clearly needs an effort to synthesize and clarify the message.

The main message of the manuscript is that IgD-BCR signal strength is reduced when compared to IgM-BCR signal strength only in vivo upon recognition of autoantigens driving differentiation (Figure 1 and MZ versus B1a cells under competitive conditions, Figure 4) or for the generation of SLPC (Figure 7). In contrast, in vitro (Figure 2), upon immunization (Figure 7) and upon recognition of auto-antigens in the *Lyn^−/−^* background (Figure 5A-B) the signals of IgD-BCR are as competitive as the signals from IgM-BCR. Taken together, these results are confusing and point out to microenvironment, cell-differentiation stage, co-receptor expression and crosstalk between receptors in a given situation as reasons modulating the signals through the BCR. To conclude that "IgD-BCR sensing antigen is impaired" is an over-interpretation of the results, a not-justified generalization and neglects a major part of the results exposed in this manuscript.

1) The whole study is based on the observation that Follicular B cells in the spleen are GFP+, indicating that they have been activated at some point during development (T2/T3 stage) by any of the receptors that induce the activation of the Nur77 locus (BCR, CD40, TLR9, TLR4…). The authors claimed already back in the first publication (Zikherman et al., 2012) that GFP positivity results from BCR signals upon encounter with autoantigens because GFP positivity is significantly reduced in BCR transgenic settings (IgHEL Tg), and again increased in the presence of antigen (IgHEL Tg/sHEL Ag Tg). However, if the encounter with autoantigens drives GFP positivity, why are there not GFP+ cells within the immature cells in the BM? At any given moment, some immature B cells are signaling through the BCR during B cell selection, one should expect a proportion of GFP+ cells in the BM. Moreover, it has been broadly demonstrated that BCR driven signals are crucial for B cell development, why are there not GFP+ B cells during B cell development? Is it possible that GFP positivity is the result of a combination of different signals happening from the T2/T3 stage on, including also BCR signals in the IgHEL Tg/sHEL Ag Tg settings? The authors discard CD40; TLR3,7,9 (*Unc93b1* mice, this information should be found at least in the figure legend); TLR7; IL4; BAFF and CXCR4 (Figure 1—figure supplement 1 and Zikherman et al., 2012). However, an obvious candidate is TLR4, no data are shown using either LPS stimulation in vitro, *TLR4^−/−^* mice, *Myd88^−/−^* mice or even better by housing the mice in pathogen free/sterile conditions. Indeed amount of mRNA of TLR4 increases from the T1 stage on to get a maximum at the Fo stage. Other receptors which levels of mRNA are upregulated from the transitional stage on are TLR6, TLR1, CD22….

2) In line with the comment before, why most of Fo cells (around 90%) are GFP+ also in the WT settings? This high percentage is expected in the IgHEL Tg/sHEL Ag Tg setting. However, in the WT situation, if GFP positivity is the consequence of autoantigen encounter, this would mean that most of B cell emigrants from the BM carry autoreactive receptors. Do the authors suggest that tolerance in mice is established mainly in the peripheral checkpoint (T2/T3)? This will be in contrast with the data in humans in which both checkpoints reduced the percentage of self-reactive BCR to half, and only 40% of new emigrants carry self-reactive BCRs, percentage that is reduced to 20% after the peripheral checkpoint.

Taken together the last two points, it seems true that autoantigens are sufficient to drive Nur77-GFP expression as claimed by the authors (IgHEL Tg/sHEL Ag Tg setting already reported in 2012), but I am not convinced at this stage that all GFP+ cells in the Fo stage are GPF+ because BCR signaling as consequence of autoantigen encounters. I rather think that GFP accumulation is the result of several activation signals from diverse receptors, including the BCR but not exclusively.

3). Because an inverse correlation between GFP levels and IgM expression is observed, the authors propose that IgM is downregulated as consequence of autoantigen encounter and therefore, activation of autoreactive B cells is prevented. The slope that correlates IgM and GFP expression in the IgM-only mice is reduced compared to the slope in WT, why is this? In other words, why in the absence of IgD, IgM cannot be optimally downregulated? In this the case also upon anti-IgM stimulation ex vivo?

In contrast to IgM, IgD levels are invariant with GFP expression, this would mean that in IgD mice GFP high cells could be easily re-activated driving autoimmunity, is this the case? The authors claim that these cells cannot be optimally activated, but the total GFP levels are the same, proving that activation was equally strong. Can IgD be downregulated upon stimulation ex vivo in the IgD-only B cells?

4) In Figure 1F, the authors showed that for a given GFP signal more IgD-BCRs are needed than IgM-BCRs, proposing that IgD-BCRs do not signal properly in vivo (supposedly, in response to antigens). Taking in consideration recent data regarding the nanoscale distribution of IgM and IgD-BCRs in protein islands, how are the levels of co-receptors such as CD19, CD22, CD20 in WT, IgM-only and IgD-only? Are these co-receptors excluded or included in the IgM protein island in the absence of IgD? Are these coreceptors modulating differently IgM-BCR and IgD-BCR signals? Differences in co-receptor levels and/or composition of these protein islands might explain why IgD stimulation in vitro, which happens solely by the BCR, is strong (Figure 2).

5) What exactly does this mean for the authors, "sensing of endogenous antigen"? or just "sensing of antigen"?

6) It is unclear to me the significance of the experiments shown in Figure 3. Both B1a and MZ B cells express preferentially IgM-BCRs. Are they generated to the same extent in IgD-only mice? If yes, is this not probing that IgD-BCR in vivo signals as efficient as IgM-BCR at least to drive differentiation of these cells? However, IgD-only B1a and M Z cells showed reduced levels of Nur77-GFP ex vivo. Again, how is the expression of co-receptors and especially of innate-like receptors in IgD-only versus WT or IgM-only B1a and MZ B cells? In these two populations all cells are GFP+ cells, it seems counterintuitive that all cells that populate the periphery are autoreactive… In the IgD-only B1a cells there is indeed a GFP- population, the authors should compare the repertoire of these GFP- cells to GFP+ cells to support they claims.

7) As cited by the authors, a link between BCR signal strength and cell fate has been established some years ago, in which weak BCR signals will allow MZ B cell generation but prevent B1 cell generation. In contrast, strong signals might favor B1 generation/maintained. No major differences were observed in B1 or MZ cell generation between IgM-only, IgD-only and WT. However, under competitive conditions, IgD-only B cells fill up preferentially the MZ B cell compartment and are unable to differentiate to B1a cells. How are these cells phenotypically? MZ are BM-derived while B1a cells are mainly fetal liver derived and long lived. It has been proposed that in B2 cells IgD provide survival signals. Is it possible that B1 cells required survival signals via IgM and that IgD cannot provide survival signals or self-renewal signals? Is it really a matter of signal strength or of signal quality? How is this piece of data related to autoantigen recognition?

8) Which bibliography supports the following sentence "endogenous auto-antigens may differ significantly from model antigens"? The authors used HEL as model antigen and as endogenous auto-antigen, is the model antigen HEL not differing from endogenous auto-antigens?

9) Similar to the results obtained in vitro, the results in the *Lyn^−/−^* mice support that both IgM and IgD-BCR provide sufficient signals for B cell activation (CD86, CD69) and for GC differentiation against the theory of the authors that "IgD-BCR sensing" of antigen is impaired.

10) Regarding the production of autoantibodies in Figure 5, why the production of anti-dsDNA antibodies is also increased in the WT cells that co-exist with the IgM-only B cells? Indeed, at 25 weeks the basal levels are elevated in all the mice compared to all the other situations. It is possible that other receptors, such as germline-encoded receptors, cooperate only with IgM-BCR to break tolerance and not with IgD? This will be independent of BCR strength but dependent of crosstalk between receptors. This crosstalk has been described at least for TLR4/IgM-BCR and RP105/IgM-BCR.

11) In Figure 6A, how are the levels of Ets1 in IgM-only? And in Figure 6B, how is intracellular IgD in WT or IgM-only mice? I do not agree with the interpretation of Figure 6E, *IgM^−/−^Lyn^−/−^* have elevated IgD serum levels when compared to *Lyn^−/−^*.

12) Why are IgD-only less competitive to enter the GC but disfavored to produced SLPC responses? Is this because BCR-strength as proposed by the authors or by the effect of isotype-specific interactions with co-receptors?

Reviewer #3:

In this manuscript the authors present an extensive body of work comparing the development and responsiveness of polyclonal B cells capable of expressing either IgM or IgD BCRs prior to undergoing class switching. Whilst there is an extensive literature comparing the functional capabilities of these two (normally) co-expressed classes of BCR, this particular study examines a variety of readouts not previously examined, in particular the apparent sensing of endogenous self-antigens. Despite some concern about the different strain origins of the original IgM and IgD KO lines, the authors appear to control for this adequately throughout the study.

Overall I found the experiments to be well-performed and thoughtfully interpreted. The data presented here go some way to resolve recent controversies in the field and as such provide a valuable source of new information to help resolve the longstanding question of why naive, resting B cells co-express IgM and IgD BCRs.

I have just a couple of specific points:

1) In the SRBC-induced responses, did the IgM KO mice possess a general deficiency in PC production apart from IgG1-switched PCs e.g. in producing unswitched PCs?

2) Similarly, was the production of switched (e.g. IgG1+) GC B cells different between the IgM-only and IgD-only B cells?

It would be important to add this information to clarify the nature of the defect involved.

[Editors’ note: what now follows is the decision letter after the authors submitted for further consideration.]

Thank you for submitting your work entitled "IgM and IgD B cell receptors differentially respond to endogenous antigens and control B cell fate" for consideration at *eLife*. Your article has been favorably evaluated by Tadatsugu Taniguchi (Senior Editor) and three reviewers, one of whom is a member of our Board of Reviewing Editors.

This manuscript provides novel insights into different functions between IgM and IgD BCRs. There are some remaining issues that need to be addressed before acceptance, as outlined below in the separate reviews. Specifically, we request that you recognize that the Nur77-GFP signals in B cells might be the result of BCR stimulation by a broader collection of non-inflammatory antigens in vivo, not only self-antigens. They might be globally named as endogenous antigens but not exclusively self-antigens, or endogenous self-antigens.

Specific comments.

Reviewer #1:

The authors have satisfactorily addressed most the reviewers’ comments and the manuscript has improved.

Reviewer #2:

The authors have satisfactorily addressed all the points I raised in my initial review.

Reviewer #3:

I have read with enthusiasm and interest the revised version of the manuscript entitled "Differential sensing of endogenous antigens and control of B cell fate by IgM and IgD B cell receptors" by Noviski et al. I appreciate the determination of the investigators to address the issues that arose during the revision process and thank the authors for their efforts.

One of my major points in the previous revision round was whether the Nur77-GFP positive B cells were indeed positive because activation of the BCR by self-antigens (as claimed by the authors) or by a combination of signals by the BCR and/or additional receptors expressed on the surface of B cells.

The authors have appropriately answered part of this question by showing that some of these receptors do not activate expression of Nur77 (TLR3, TLR7, TLR9, CXCR4). Other receptors, such as TLR4, TLR9 and TLR1/2 do activate expression of Nur77-GFP (Figure Author response image 6). However, the authors demonstrate here (Figure 1—figure supplement 3) that upregulation of Nur77-GFP by these receptors is equal between IgM-only and IgD-only B cells (although the concentration of ligands chosen are too high). The authors also show in the new version, that endogenous levels of Nur77 transcripts in B cells are equally elevated between Germ-free mice and SPF, arguing that Nur77 signals in B cells do not from microbial stimulation (although these signals might come from the endotoxin present in the chow, or did the authors used endotoxin-free chow?). Moreover, the levels of endogenous Nur77 transcripts were equally elevated between MyD88KO and MyD88WT peritoneal B1a cells. Here, I would like to ask why peritoneal B1a cells were chosen and not splenic B cells as in Figure 1—figure supplement 1D-E?

However, I do not agree that the Nur77-GFP signals in B cells are the consequence of self-antigen (self-antigen defined by an antigen encoded by the DNA of the self-organism) stimulation by the BCR. I disagree with the authors using self-antigen (subsection “Cell-intrinsic skewing of B cell development by the IgM and IgD BCRs”, first paragraph), endogenous auto-antigen (subsection “Either IgD or IgM BCR is sufficient to mediate polyclonal B cell activation and germinal center differentiation in *Lyn^−/−^* mice”, first paragraph) and antigen encounters in vivo (subsection “IgD BCRs sense endogenous antigens less efficiently than IgM BCRs”) as synonymous. I fully agree that Nur77-GFP signals are upregulated upon BCR stimulation by antigen and that antigen is necessary to drive Nur77-GFP expression in vivo. But I do not agree that self-antigen is necessary to drive Nur77-GFP expression in vivo. My arguments are:

• Nur77-GFP signals are highly upregulated when B cells leave the sterile/defined microenvironment of the BM. Nur77-GFP expression timely occurs when cells encounter "antigens" in the periphery, and those antigens activate the BCR. But these antigens are not necessarily self-antigens, they can be food-antigens, LPS (from the chow, antibodies against LPS are broadly recognized), and other non-pathological or non-inflammatory antigens recognized by the BCR. These antigens might activate B cells to upregulate Nur77, but in the absence of T cell help or an inflammatory milieu will not drive an immune response.

• The results with the IgHEL BCR Tg mice (Author response image 1) are fully compatible with the above-described scenario. These mice express a clonal BCR not able to recognize any self-antigen, but also, not able to recognize any of the possible non-inflammatory antigens that will be encountered in the periphery, for this reason the signal of Nur77 is lower. This interpretation is also compatible with higher level of Nur77-GFP in HEL Tg cells exposed to cognate HEL antigen that in WT. Related to this, the results show in Author response image 6 are also compatible with this scenario since Nur77-GFP upregulation is reduced in IgHEL BCR Tg mice in respond to LPS (small reduction), CpG (clear at lower concentration), and to Pam3CSK4 (visible at both concentrations). A possible interpretation of these results is that the HEL Tg BCR is not able to recognize epitopes in those substances that, together with the corresponding TLR, aid to activate Nur77-GFP in B cells.

• The results with the VH3H9 Tg mice (Author response image Figure 3) are also fully compatible with the above-described scenario, since the κ+ cells (not recognizing endogenous DNA) also up-regulate (even to the same levels than λ+ cells) Nur-77 upon leaving the BM. Most probably because they encounter non-inflammatory antigens in the periphery.

Taken together, I fully agree that the present manuscript is improved upon revision and most of my requests have been clarify. However, I still request that the authors recognize that the Nur77-GFP signals in B cells might be the result of BCR stimulation by a broader collection of non-inflammatory antigens in vivo, not only self-antigens. They might be globally named as endogenous antigens but not exclusively self-antigens, or endogenous self-antigens. The main conclusion of the manuscript is unaltered, namely that IgD-BCR signals differently upon encounter of non-inflammatory antigens in vivo, but to narrow this conclusion only to self-antigens is not proven and not justified.

---

## [Author Response]

[Editors’ note: the author responses to the first round of peer review follow.]

Reviewer #2:The manuscript entitled "Differential sensing of endogenous antigens and control of B cell fate by IgM and IgD B cell receptors" by Noviski et al. follows a series of manuscripts started back in 2012 with the publication of the Nur77-GFP mice (Zikherman et al., 2012). Nur77-GFP mice have thus emerged as a useful tool to study B cell antigen receptor (BCR) and T cell antigen receptor (TCR) signaling in vivo.In the present manuscript, the authors aimed to answer the long-standing question why mature B cells express two BCR isotypes, IgM and IgD, with identical antigen-specificity, and how these two isotypes impose B cell tolerance in vivo.A major issue for me is the clarity of the present manuscript. Although the initial question is clear and well defined, a good rational flow along the experiments, figures and the text is missing.

We have made manuscript edits to clarify the flow of logic in our Introduction (last paragraph), and to transition more smoothly between sections throughout the manuscript.

The structure of the paper reflects the nature of the question we sought to address, namely differences between the response of the IgM and IgD BCRs to a diverse repertoire of bona fide endogenous antigens in vivo. Rigorously studying in vivo antigen sensing in the context of a polyclonal repertoire requires examining a broad array of BCR-dependent physiological readouts. The paper begins by establishing the specificity of the Nur77-eGFP reporter as a read-out of endogenous antigen-dependent signaling in B cells (below in the rebuttal we review new data establishing specificity and validity of this reporter). We next introduce this reporter into mice lacking either IgM or IgD to determine how endogenous antigens-dependent signals are transduced by these BCR isotypes. We observe that IgD drives less reporter expression per receptor than IgM, and we proceed to rule out differences in downstream signal transduction as an explanation. Next, we observe that signaling in response to residual endogenous antigens is impaired in IgD-only cells (as read out by basal calcium and ex vivo GFP induction). Indeed, we show that marginal zone (MZ) and B1a cells that express IgD but lack IgM induce less Nur77-eGFP than WT and this is not attributable to repertoire differences. Subsequent figures examine sensing of endogenous antigens in relevant in vivo contexts: B cell development and autoimmune disease. In the context of autoimmune disease, we identified a defect in rapid plasma cell responses by IgD-only cells. The final figure builds upon this finding, and identifies a major defect in rapid IgG1^+^ plasma cell induction by IgD-only cells in the context of immunization with T-dependent antigens, but intact generation of germinal center responses.

Also the way the data are presented is highly confusing and prevents comparison between figures.

We anticipated that the use of IgM^−/−^, IgD^−/−^, and IgM^−/−^IgD^−/−^ mice (containing mixed populations of IgM-only, WT, and IgD-only B cells) along with IgM^−/−^, WT, and IgD^−/−^ mice might be challenging to keep track of. For this reason, we intentionally generated a consistent color scheme to aid in interpretation of data, and we provided schematic diagrams to clarify our logic (Figure 6—figure supplement 2, Figure 7—figure supplement 2). We labeled all graphs with both the genotype of the mouse, and the “genotype” of the B cells specifically to aid in clarity. In general, cells that express IgM alone are blue and cells that express IgD alone are red (WT cells are grey or orange depending on allotype). We have revised our figures to remove as many deviations from this color scheme as possible, specifically in Figure 5E, F, Figure 7B, D, Figure 7—figure supplement 1B, C, D.

The manuscript clearly needs an effort to synthesize and clarify the message.The main message of the manuscript is that IgD-BCR signal strength is reduced when compared to IgM-BCR signal strength only in vivo upon recognition of autoantigens driving differentiation (Figure 1 and MZ versus B1a cells under competitive conditions, Figure 4) or for the generation of SLPC (Figure 7).

Yes, this message is the one we hoped to communicate. Our evidence for this is also grounded in Manuscript Figure 3 demonstrating reduced Nur77-eGFP expression in IgD-only relative to IgM-only MZ and B1a B cells. We also show defect in SLPC generation and autoantibody production by IgD-only cells in response to endogenous antigens in the *Lyn^−/−^* model (Figures 5, 6). So, in sum, Figures 3, 4, 5, 6, and 7 provide evidence to support this message.

In contrast, in vitro (Figure 2), upon immunization (Figure 7) and upon recognition of auto-antigens in the Lyn^−/−^ background (Figure 5A-B) the signals of IgD-BCR are as competitive as the signals from IgM-BCR.

This is in part an error of interpretation. We showed (Figure 2.) that in vitro IgD was equally if not more efficient at signal transduction than IgM in response to antibody cross-linking, in contrast to our in vivo observations. We suggest that coupling of the IgD BCR to downstream signaling machinery is intact but sensitivity to endogenous antigens is specifically impaired. We reconcile these results by suggesting that responses to endogenous antigens differ from those triggered by antibody-mediated crosslinking of the receptors in vitro. In response to endogenous antigens, we show that polyclonal B cell activation and GC B cell fate are intact, while autoantibody and SLPC generation are defective (Figures 5, 6). In response to T-dependent immunizations, we show impaired IgG1 SLPC generation, but intact GC B cell fate (Figure 7). We reconcile these observations by suggesting that the stringent requirement for strong BCR signaling for SLPC (but not GC) generation accounts for these observations.

Taken together, these results are confusing and point out to microenvironment, cell-differentiation stage, co-receptor expression and crosstalk between receptors in a given situation as reasons modulating the signals through the BCR. To conclude that "IgD-BCR sensing antigen is impaired" is an over-interpretation of the results, a not-justified generalization and neglects a major part of the results exposed in this manuscript.

We examine a wide variety of readouts known to be affected by BCR signal strength (ex vivo calcium, ex vivo GFP, in vivo GFP, MZ/B1 fate, SLPC and GC fate in Lyn^−/−^, SLPC and GC fate in response to immunization), and we cite literature describing how BCR signal strength affects the propensity of a B cell to adopt each fate. The common thread is that IgD-only cells are inefficient at adopting the fates that require the strongest in vivo BCR signaling in response to endogenous antigens. It is worth emphasizing that IgM and IgD are BCR isotypes that signal in response to antigen. Therefore, it should not be extremely controversial to propose in vivo antigen-dependent signaling differences as an explanation for phenotypes in mice expressing each isotype in isolation.

We acknowledge that complex in vivo phenotypes may reflect cooperation between IgM or IgD BCRs and some to-be-defined signal, but it is not possible or reasonable to exclude every known receptor expressed on B cells in this manuscript. That said, we provide important new data excluding a role for microbial stimulation of pattern-recognition receptors as well as TLR-MyD88-dependent signaling in regulating Nur77 expression in vivo by using two independent approaches (germ free mice and conditional deletion of MyD88 in B cells, Figure 1—figure supplement 1D, E, F). Table 1 lists data to exclude a role for: commensal flora and signaling downstream of MyD88, endocytic TLRs, Jak-Stat, CXCR4, CD40, and BAFF in the regulation of Nur77 expression in B cells in vivo under steady-state conditions. We further show in new data that a range of TLR ligands do not specifically require either IgM or IgD to signal and therefore would not account for our phenotypes (Figure 1—figure supplement 3). In the fifth paragraph of our revised Discussion, we explicitly state that the most likely mechanistic explanation for our findings are structural differences between the IgM and IgD isotypes, which could either directly influence transduction of endogenous antigen-driven signals or influence pairing with and access to relevant co-receptors such as CD19 and CD22 that in turn modulate BCR signaling.

We have no wish to over-interpret our data, but one cannot expect a model that synthesizes the data (clarity of message as requested above) without any interpretation or conclusions. Some degree of interpretation is necessary to help the reader make sense of a complex set of in vivo phenotypes. We have made every effort in our revised manuscript to make the clear distinction between description and interpretation of data.

Our manuscript should also be read against a background literature which has failed to identify significant non-redundant functions of these co-expressed BCR isotypes in vivo. We have done so for the first time and present a compelling model to explain why IgM and IgD are co-expressed on naïve B cells, and how down-regulation of the former but not the latter on self-reactive cells could function as a vital tolerance mechanism.

1) The whole study is based on the observation that Follicular B cells in the spleen are GFP+, indicating that they have been activated at some point during development (T2/T3 stage) by any of the receptors that induce the activation of the Nur77 locus (BCR, CD40, TLR9, TLR4…). The authors claimed already back in the first publication (Zikherman et al., 2012) that GFP positivity results from BCR signals upon encounter with autoantigens because GFP positivity is significantly reduced in BCR transgenic settings (IgHEL Tg), and again increased in the presence of antigen (IgHEL Tg/sHEL Ag Tg).

We make use of the Nur77-eGFP reporter in our manuscript precisely because our goal is to understand response of a polyclonal repertoire of IgM and IgD isotypes to endogenous antigens in vivo (rather than to model antigens). Major points 1 and 2 raised by this reviewer question whether Nur77-eGFP is a valid reporter of antigen-dependent signaling by B cells in vivo. We argue that this degree of skepticism is not justified. We provide data in this rebuttal and in our revised manuscript to further validate this reporter precisely because it aids in interpretation of our cellular phenotypes. As the reviewer notes and we have shown genetically in previous work (Zikherman et al., 2012), endogenous antigen is necessary and sufficient for Nur77-eGFP expression in vivo in mature B cells under steady state conditions (Manuscript Figure 1A, Author response image Figure 1).

**Author response image 1. respfig1:** Antigen is necessary and sufficient for Nur77-eGFP expression in B cells; IgHEL-sHEL interaction drives higher Nur77-eGFP expression than the wild type BCR repertoire. Panel (**A**) take from Manuscript Figure 1A. Panel (**B**) depicts quantification of mean Nur77-eGFP MFI -/- SEM in N=5 biological replicates.

In our rebuttal and revised manuscript, we present substantial new data to exclude the contribution of other immunoreceptor pathways as regulators of Nur77-eGFP expression in B cells under steady-state conditions, especially those mediated by microbial stimuli. We have included a new table that summarizes all published and new figure panels that address specificity of reporter expression in B cells in response to immunoreceptor signals in vitro and in vivo (Table 1). These data collectively demonstrate that in vivo expression of Nur77-eGFP in B cells under steady state conditions is specifically regulated by antigen and regulators of BCR signaling (e.g. CD45, Lyn), but not by other immunoreceptors (including each receptor listed by the reviewer above: CD40, TLR9, TLR4) and therefore represents a VALID readout of antigen-dependent BCR signaling in vivo. We now proceed to address specific reviewer questions about the function of this reporter in our point-by-point response below.

However, if the encounter with autoantigens drives GFP positivity, why are there not GFP+ cells within the immature cells in the BM? At any given moment, some immature B cells are signaling through the BCR during B cell selection, one should expect a proportion of GFP+ cells in the BM. Moreover, it has been broadly demonstrated that BCR driven signals are crucial for B cell development, why are there not GFP+ B cells during B cell development?

We believe the explanation for low levels of Nur77-eGFP upregulation at early stages of B cell development is two-fold:

1) Immature B cells are more refractory to BCR-induced Nur77 upregulation than transitional and mature B cells. Ex vivo stimulation of reporter B cells at sequential stages of development shows that immature BM B cells are less responsive to anti-IgM stimulation than more mature B cell subsets (Author response image 2). This may be due in part to differences in receptor expression (lower levels of surface IgM, absence of IgD), but may also reflect differences in signaling machinery or chromatin landscape. Importantly, we see identical phenomenology comparing two independently generated Nur77-eGFP BAC Tg reporters (Moran et al., 2011; Zikherman et al., 2012).

**Author response image 2. respfig2:** In vitro anti-IgM stimulation upregulates Nur77-eGFP expression in a stage-specific manner. BM and splenic B cells from our Nur77-eGFP reporter mice (Zikherman et al., 2012) as well as independent reporter line (Moran et al. JEM 2011) were stimulated in vitro with media alone (shaded gray histogram) or 10μg/ml anti-IgM (solid black line) for 3 hours, and subsequently stained to detect subset markers as well as GFP.

2) Highly self-reactive BM B cells are efficiently silenced via receptor editing and deletion. Indeed, when we force expression of a highly self-reactive BCR Tg, we observe that immature BM B cells in fact do upregulate Nur77-eGFP expression (Author response image 3). V_H_3H9 is a heavy chain (HC) originally cloned from the MRL/lpr lupus mouse model and biases the entire BCR repertoire towards DNA-reactivity, particularly when paired with endogenous V_λ_1 light chain (LC) (Erikson et al., 1991). This HC has been knocked-in to the endogenous BCR locus to generate site directed (sd) V_H_3H9 Tg mice in which the majority of B cells express this HC paired with endogenous LCs. This model system is extremely powerful because bona-fide DNA-reactive B cells can be tracked through development in a polyclonal repertoire on the basis of surface V_λ_1 expression. B cells expressing dsDNA-specific V_λ_1 BCRs persist in the periphery in sd-V_H_3H9 mice despite extensive receptor editing and counter-selection, in part because V_λ_1 light chain expression is itself the result of extensive receptor editing and “traps” self-reactivity in the repertoire (Erikson et al., 1991; Fields et al., 2005). We have generated reporter mice harboring the site-directed (sd) V_H_3H9 Tg in the endogenous BCR locus. dsDNA-reactive V_λ_1 B cells turn on GFP early in development and express higher levels in maturity, consistent with their chronic self-antigen stimulation (Author response image 3, manuscript in preparation from our laboratory). These data support the Nur77-eGFP reporter as a sensitive marker of B cell self-reactivity, and argue that highly self-reactive B cells do upregulate reporter expression during BM development, but these cells normally represent a small proportion of the repertoire and are efficiently eliminated in WT mice.

Because these points are an important response to the reviewer query about the sensitivity of the Nur77-eGFP reporter to antigen-dependent signaling during B cell development, but are not directly relevant to the manuscript under review, we present these data in the rebuttal, but have not incorporated them into the revised manuscript. Moreover, data in Author response image 1 and Author response image 3 are taken from a manuscript in preparation in our laboratory.

**Author response image 3. respfig3:** Self-reactive B cells upregulate Nur77-eGFP at early stages of B cell development. Nur77-eGFP BAC Tg was crossed to the Vh3H9 HC site directed Tg. BM and splenic cells from these mice (red) and unrestricted repertoire reporter mice (gray) were harvested and stained ex vivo to identify B cell subsets as well as surface light chain expression. Histograms depicting GFP expression in K+ and L+ B cells subsets are representative of > 9 biological replicates.

Is it possible that GFP positivity is the result of a combination of different signals happening from the T2/T3 stage on, including also BCR signals in the IgHEL Tg/sHEL Ag Tg settings? The authors discard CD40; TLR3,7,9 (Unc93b1 mice, this information should be found at least in the figure legend); TLR7; IL4; BAFF and CXCR4 (Figure 1—figure supplement 1 and Zikherman et al., 2012). However, an obvious candidate is TLR4, no data are shown using either LPS stimulation in vitro, TLR4^−/−^ mice, Myd88^−/−^ mice or even better by housing the mice in pathogen free/sterile conditions. Indeed amount of mRNA of TLR4 increases from the T1 stage on to get a maximum at the Fo stage. Other receptors which levels of mRNA are upregulated from the transitional stage on are TLR6, TLR1, CD22….

We have modified the figure legend.

As suggested by this reviewer, we have directly addressed the role of microbial stimulation of TLR pathways in regulating Nur77 expression in B cells in vivo at steady-state.

1) Specifically, our revised manuscript examines both germ-free mice and mice with conditional deletion of MyD88 in B cells (MB1cre MyD88 fl/fl mice) in order to demonstrate that Nur77 expression is INDEPENDENT of commensal flora and TLR /MyD88-dependent signaling pathway at steady-state in vivo. Since re-deriving reporter mice under germ-free conditions and breeding the reporter into the MyD88 conditional background would each be extremely time consuming and expensive, we instead assayed endogenous Nur77 protein in B cells and endogenous *Nr4a1* transcript in splenocytes from these mice. We first took advantage of *Nr4a1^−/−^* mice to confirm specificity of intracellular staining for endogenous Nur77 protein (Author response image 4, Figure 1—figure supplement 1D, F). Because qPCR primer sets to detect Nr4a1 expression flank a neo cassette in *Nr4a1^−/−^* mice, they also serve as a specificity control for *Nr4a1* primer sets (Figure 1—figure supplement 1E). We next verified that endogenous Nur77 protein and Nr4a1 transcript serve as sensitive measures of Nur77-eGFP reporter expression in B cells under steady state conditions (Author response image 4). We demonstrate that neither endogenous Nur77 protein nor endogenous *Nr4a1* transcript are upregulated under steady state conditions in splenic B cells / splenocytes in SPF mice relative to germ-free mice (Figure 1—figure supplement 1D, E). We further show that endogenous Nur77 protein levels are equivalent in WT and MB1-cre MyD88 fl/fl B1a cells (Figure 1—figure supplement 1F). We also confirm successful deletion of MyD88 in B cells in these mice (Author response image 5).

**Author response image 4. respfig4:** Endogenous *Nr4a1* transcript and Nur77 protein correlate with Nur77-eGFP reporter expression in B cells. (**A**) Intracellular staining of endogenous Nur77 protein in splenic B cells from Nr4a1+/+, -/-, and -/- mice. Graph depicts mean -/- SEM of N=3 mice / genotype. (**B**) Nur77-eGFP reporter B cells were stained to detect endogenous Nur77 protein intra-cellularly. Graph depicts mean MFI of endogenous Nur77 in splenic B cells with low or high GFP expression (15% lowest and 15% highest) -/- SEM of N=3 mice. (**C**) Mature naive reporter B cells were sorted for 3% highest and 3% lowest GFP expression and subjected to RNASeq. Graph depicts mean FPKM of Nr4a1 transcripts -/- SEM of N=3 biological replicates.

**Author response image 5. respfig5:** Efficient conditional deletion of MyD88 in MB1-Cre MyD88 fl/fl B cells (“MyD88 cKO”). Splenocytes from MyD88 cKO or control mice were stimulated for 2 hours with LPS (333 ng/ml). Graph depicts mean MFI of Intracellular staining to detect Nur77 upregulation in splenic B cells -/- SEM in N=4 mice. Nur77-deficient mice are a staining control.

2) In our Nature 2012 paper (Figure 3) and in Figure 1A of this manuscript we show that forced expression of the IgHEL BCR Tg in the absence of endogenous cognate antigen nearly eliminates all Nur77-eGFP expression in resting naïve splenic B cells (Author response image 1). This demonstrates that antigen is *necessary* to drive Nur77eGFP expression in vivo. We present new data to show that this loss of Nur77-eGFP expression is not due to impaired TLR-dependent signaling in these B cells as they are fully competent to respond to TLR ligands in vitro (Author response image 6). Together this argues that since microbial sensing by IgHEL B cells is intact, and yet Nur77-eGFP expression is lost, endogenous antigen is necessary for Nur77-eGFP expression, and Nur77-eGFP expression reflects endogenous antigen rather than indirect loss of a TLR-dependent signaling pathway.

**Author response image 6. respfig6:** Naïve B cells are competent to upregulate Nur77-eGFP in response to canonical TLR ligands even in the absence of endogenous antigen. Splenocytes from non-BCR Tg and IgHEL BCR Tg reporter mice were stimulated overnight with low and high doses of ligands for TLR4 / rp105 (**A**), TLR9 (**B**), and TLRs1/2 (**C**). Histograms depict GFP fluorescence in B220+ splenocytes. As previously reported, Nur77-eGFP expression is low in the absence of endogenous antigen (IgHEL BCR Tg), but these cells are nevertheless responsive to a broad range of TLR ligands.

3) Lastly, in Figure 1—figure supplement 3A, we compare TLR-dependent signal transduction by IgM-deficient and IgDdeficient B cells in vitro and show that signaling downstream of LPS, CpG, and Pam3CSK4 through TLR4/Rp150, TLR9, and TLRs1/2 respectively are not dependent upon either the IgM or the IgD BCR. This allows us to conclude that differences between IgM-deficient and IgD-deficient B cells in vivo are not due to defective TLR-dependent signaling.

Taken together, data presented here and summarized in Table 1 supports specificity and validity of Nur77-eGFP reporter as a marker of endogenous antigen-dependent signaling in vivo under steady state conditions.

We discuss CD22 separately below.

2) In line with the comment before, why most of Fo cells (around 90%) are GFP+ also in the WT settings? This high percentage is expected in the IgHEL Tg/sHEL Ag Tg setting. However, in the WT situation, if GFP positivity is the consequence of autoantigen encounter, this would mean that most of B cell emigrants from the BM carry autoreactive receptors. Do the authors suggest that tolerance in mice is established mainly in the peripheral checkpoint (T2/T3)? This will be in contrast with the data in humans in which both checkpoints reduced the percentage of self-reactive BCR to half, and only 40% of new emigrants carry self-reactive BCRs, percentage that is reduced to 20% after the peripheral checkpoint.Taken together the last two points, it seems true that autoantigens are sufficient to drive Nur77-GFP expression as claimed by the authors (IgHEL Tg/sHEL Ag Tg setting already reported in 2012), but I am not convinced at this stage that all GFP+ cells in the Fo stage are GPF+ because BCR signaling as consequence of autoantigen encounters. I rather think that GFP accumulation is the result of several activation signals from diverse receptors, including the BCR but not exclusively.

We have indeed proposed that most mature B cells harbor self-reactivity to endogenous antigens (albeit mild) that is proportional to Nur77-eGFP reporter expression, and that Nur77-eGFP expression in those cells is a result of endogenous antigen recognition. This was a central part of the message of our Nature 2012 paper. Our evidence in support of this contention is three-fold:

a) Restricting the normal diverse BCR repertoire to force recognition of a non-murine antigen (via IgHEL BCR Tg) and thereby eliminate endogenous antigen recognition results in near-complete loss of Nur77-eGFP expression in B cells in vivo (Figure 1A; Zikherman et al., 2012: Figure 3B, C, Author response image 1). Notably, this occurs despite the fact that surface BCR expression in IgHEL BCR Tg cells is higher than that found on WT B cells, and despite the fact that these cells are competent to respond to TLR ligands (Author response image 6). Reintroducing model antigen (sHEL Tg) genetically into this system is sufficient to upregulate Nur77-eGFP in B cells and recapitulate GFP expression seen in a normal diverse mature BCR repertoire. Importantly, the GFP MFI seen in IgHEL BCR Tg B cells exposed to high affinity cognate antigen sHEL in vivo is *considerably higher* than that seen in WT B cells with a diverse repertoire (Author response image 1). This suggests that WT B cells *do* see endogenous antigens, but at considerably lower intensity / affinity than that encoded by the IgHEL / sHEL model.

b) We have shown in previously published work that Nur77-eGFP B cells harbor BCRs with higher ANA-reactivity as measured by ELISA (Zikherman et al., 2012: Figure 4E). Similar findings of autoreactivity among IgM^lo^ mature B cells from mice and humans (corresponding to B cells with higher GFP and endogenous Nur77 expression) have been reported by independent groups in mice and humans (Kirchenbaum et al., 2014; Quach et al., 2011).

c) Moreover, in published data and the present revised manuscript and rebuttal, we systematically exclude a contribution by: commensal microbes, MyD88-dependent signaling, CD40-dependent signaling, CXCR4-dependent signaling, Jak-Statdependent signaling, and BAFF-dependent signals (see Table 1, Figure 1—figure supplement 1).

The reviewer wishes to reconcile our argument that Nur77-eGFP expression in naïve mature B cells reflects reactivity to endogenous antigens with data reported by Wardemann, Nussenzweig and colleagues (Wardemann et al., 2003). Wardemann at al. show that anti-nuclear-reactivity of mature human B cells measured by immunofluorescence assay (IFA) are seen in only about 20% of cells. This study of necessity uses an arbitrary cutoff and an artificial readout (ANA IFA) to identify “autoreactive B cells”. Our readout (GFP fluorescence) is a continuous, linear, and sensitive reporter of bona fide reactivity to endogenous antigens without any arbitrary cutoff other than detection of fluorescence by flow cytometer. We propose that the degree of endogenous antigen recognition by the naïve mature repertoire is “weak / mild” in comparison to genetic models of B cell autoreactivity (see Author response image 1, Author response image 3), but is detectable by the Nur77-eGFP reporter and is above the baseline seen with a bona fide non-self-reactive BCR (IgHEL Tg). The self-reactivity detected by ANA IFA used by the Nussenzweig laboratory may be distinct from true endogenous antigen recognition and/or this assay may have a lower sensitivity for detection of self-reactivity. Indeed, Ignacio Sanz and others have characterized a naturally-occurring human BCR heavy chain (VH34-4) that recognizes an epitope on the surface of RBCs that would presumably not be detected by an ANA assay (Quach et al., 2011; Reed et al., 2016).

Indeed, there is an extensive B cell literature that is consistent with our contention that self-reactivity (mild) is a feature of the normal mature B cell repertoire, and may actually be selected for. Prior work has supplied indirect evidence for antigen-dependent positive selection of B cells (Cancro and Kearney, 2004; Cariappa and Pillai, 2002; Rosado and Freitas, 1998; Su et al., 2004; Thomas et al., 2006; Wang and Clarke, 2003). Studies have shown that self-ligands can drive selection of B cells into the mature primary immune repertoire in the context of BCR Tg models (Hayakawa et al., 2003; Wang and Clarke, 2003). Analysis of AgR repertoire diversity has revealed a restricted repertoire in mature relative to immature B cells, suggesting active selection by antigen (Freitas et al., 1989; Gu et al., 1991; Levine et al., 2000).

3). Because an inverse correlation between GFP levels and IgM expression is observed, the authors propose that IgM is downregulated as consequence of autoantigen encounter and therefore, activation of autoreactive B cells is prevented. The slope that correlates IgM and GFP expression in the IgM-only mice is reduced compared to the slope in WT, why is this? In other words, why in the absence of IgD, IgM cannot be optimally downregulated? In this the case also upon anti-IgM stimulation ex vivo?

We have explicitly addressed this point in our revised manuscript (subsection “Dual expression of IgM and IgD BCRs is required to establish a broad dynamic range of BCR responsiveness across the repertoire”, last paragraph; Discussion, eighth paragraph). Here we clarify further: previous work from the Rajewsky lab identified a critical survival function for the BCR, as B cells die upon acute deletion of the BCR (Kraus et al., 2004; Lam et al., 1997). Moreover, recent work from Goodnow lab describes a novel ENU-generated mutant that lacks IgD expression but does not compensatorily upregulate surface IgM (dmit mutant), in contrast to the IgD^−/−^ mice studied in our manuscript (Sabouri et al., 2016). The dmit mutant B cells have a major defect in survival, suggesting that not only is BCR expression required for B cell survival, but the amount matters. In Figure 1D, we show that IgM-only (IgD^−/−^) B cells have 60% as much total receptor as WT cells. Since these IgM-only cells express just one receptor, any downregulation in IgM leads to an equivalent downregulation in total surface receptor. WT cells express two isotypes, and we propose that the high level of IgD on all WT cells gives them “room” to downregulate IgM more extensively without losing BCR-mediated survival. We propose here and in our manuscript that IgM downregulation in IgD^−/−^ mice is limited by survival of such cells. It is not possible to accurately recapitulate tonic survival function of the BCR in vitro.

In contrast to IgM, IgD levels are invariant with GFP expression, this would mean that in IgD mice GFP high cells could be easily re-activated driving autoimmunity, is this the case?

This would be the case *if* IgD BCR were sensitive to endogenous antigen stimulation. Our manuscript argues that IgD is less sensitive than IgM to endogenous antigens. We propose that IgD’s reduced sensitivity towards self-antigens permits high surface expression on autoreactive cells (thereby facilitating their survival) without promoting aberrant activation in response to endogenous antigens. As IgD can respond robustly to anti-kappa stimulation in vitro, we propose that IgD is maintained on autoreactive cells to permit both survival of these cells and to allow for activation upon encounter with exogenous antigens. Our data in Figure 7 show that this subsequent activation through IgD has limits, particularly with regard to IgG1^+^ PC generation. The Goodnow lab has proposed (Model in Figure 7—figure supplement 2) that promotion of GC fate in this context allows autoreactive IgD^hi^IgM^lo^GFP^hi^ cells to mutate away from self-reactivity(Reed et al., 2016; Sabouri et al., 2016; Sabouri et al., 2014).

The authors claim that these cells cannot be optimally activated, but the total GFP levels are the same, proving that activation was equally strong.

GFP levels are the same, but it takes twice as much IgD BCR than IgM BCR to achieve comparable Nur77-eGFP expression (Figures 1B, D, E, F). Moreover, IgD-only B1a and MZ B cells have higher BCR expression but lower GFP expression than IgM-only comparators (Figure 3, Figure 3—figure supplement 1). Furthermore, when we reduce selection pressures on the MZ repertoire (BAFF Tg), resulting in comparable surface expression of IgM and IgD, we unmask a markedly reduced expression of Nur77-eGFP in IgD-only MZ B cells (Figure 3D, Figure 3—figure supplement 1A). In most of our manuscript, we focus on BCR-dependent phenotypes that are present *in spite* of high surface IgD receptor expression.

Can IgD be downregulated upon stimulation ex vivo in the IgD-only B cells?

IgD can be downregulated on the surface of both WT and IgD-only cells in response to anti-IgD or anti-kappa stimulation. In preliminary studies, we didn’t observe significant differences between the isotypes with regards to receptor downregulation in response to anti-kappa stimulation, so we did not pursue this further.

4) In Figure 1F, the authors showed that for a given GFP signal more IgD-BCRs are needed than IgM-BCRs, proposing that IgD-BCRs do not signal properly in vivo (supposedly, in response to antigens). Taking in consideration recent data regarding the nanoscale distribution of IgM and IgD-BCRs in protein islands, how are the levels of co-receptors such as CD19, CD22, CD20 in WT, IgM-only and IgD-only? Are these co-receptors excluded or included in the IgM protein island in the absence of IgD? Are these coreceptors modulating differently IgM-BCR and IgD-BCR signals? Differences in co-receptor levels and/or composition of these protein islands might explain why IgD stimulation in vitro, which happens solely by the BCR, is strong (Figure 2).

We agree that differential association with B cell coreceptors that directly modulate BCR signal transduction (e.g. CD19, CD22) may contribute to differential signaling by IgM and IgD. Our revised Discussion incorporates this possibility (fifth paragraph). We studied surface expression of CD22, CD19, and CD45 on IgM-only, IgD-only, and WT B cells via FACS staining and found no differences (Author response image 7), but we did not assess receptor co-localization by microscopy. We believe that further exploring this possibility is beyond the scope of the current manuscript, but it is a fruitful avenue to pursue in the future.

**Author response image 7. respfig7:** BCR, co-receptor and CD45 expression on IgM or IgD-deficient B cells. Splenocytes from IgM-/-, IgD-/-, or WT ice were stained to detect surface expression of CD19, CD22, CD45, CD79b, and kappa light chain. Histograms depict CD23+ splenocyte gate. Differences in surface kappa expression between genotypes consistent with Figure 1D, Figure 1—figure supplement 2B, C.

5) What exactly does this mean for the authors, "sensing of endogenous antigen"? or just "sensing of antigen"?

When referring to antigen sensing, we mean that the BCR not only binds to an antigen but also transmits a biochemical signal (can be detected by basal calcium tone, by Nur77-eGFP expression). The B cell then evaluates this signal and can either remain quiescent or undergo respond (proliferation, differentiation, etc.) “Sensing of endogenous antigen” refers to the BCR sensing an antigen in the absence of infection or immunization. As we describe above, the relevant signals in the populations we study are not microbial in origin. We have clarified our use of this term in the Introduction of our revised manuscript (last paragraph).

6) It is unclear to me the significance of the experiments shown in Figure 3. Both B1a and MZ B cells express preferentially IgM-BCRs. Are they generated to the same extent in IgD-only mice? If yes, is this not probing that IgD-BCR in vivo signals as efficient as IgM-BCR at least to drive differentiation of these cells? However, IgD-only B1a and MZ cells showed reduced levels of Nur77-GFP ex vivo.

As shown in Figure 4—figure supplement 1, IgD-only mice generate significantly fewer B1a cells and more MZ cells than WT and IgM-only mice. This altered B cell fate is markedly exaggerated in a competitive setting (Figure 4). Extensive and cited literature suggests that MZ fate requires weak BCR signals while B1a cell fate requires strong BCR signals. Thus, we demonstrate that IgD-only B cells exhibit impaired BCR signaling in response to endogenous antigens in vivo (Figure 3), and exhibit B cell fates consistent with this (Figure 4). Since signal strength has been shown to be important in the development of these subsets, we use this result as support for our claim that IgD senses endogenous antigens more weakly than IgM does.

Again, how is the expression of co-receptors and especially of innate-like receptors in IgD-only versus WT or IgM-only B1a and MZ B cells?

Again, how is the expression of co-receptors and especially of innate-like receptors in IgD-only versus WT or IgM-only B1a and MZ B cells?

The phenotype of B1a and MZ B cells from IgD-only, IgM-only and WT mice is grossly normal, as is coreceptor expression (Author response image 8). We have not examined surface expression of TLRs on these cell populations but we demonstrate that splenic B cell Nur77-eGFP expression is unaffected by commensal flora, and that MyD88-deficiency does not alter Nur77 expression in B1a cells (new Figure 1—figure supplement 1D, E, F).

**Author response image 8. respfig8:** Grossly normal surface phenotype of B1a and MZ B cells in WT, IgM-/- and IgD-/- mice. (**A**) Plots show CD19+ PerC cells gated to identify CD5+ B1a cells (reduced in IgM-/- as shown in Figure 4—figure supplement 1). Histograms depict CD5 and CD19 expression in B1a cells. (**B**) Plots show B220+ splenocytes gated to identify CD21hiCD23lo MZ B cells. Histograms depict CD21 and CD23 expression in MZ B cells.

In these two populations all cells are GFP+ cells, it seems counterintuitive that all cells that populate the periphery are autoreactive. In the IgD-only B1a cells there is indeed a GFP- population, the authors should compare the repertoire of these GFP- cells to GFP+ cells to support they claims.

The B1a B cell receptor (BCR) repertoire is highly restricted and selected on the basis of germ-line-encoded reactivity to self-antigens such as phosphatidylcholine (PtC), which is exposed on the membranes of dying cells (Baumgarth, 2016; Bendelac et al., 2001; Yang et al., 2015). Indeed, the Herzenberg lab has recently shown that a small number of PtCspecific CDR3s represent over a third of the entire B1a repertoire implying active selection for this specificity (Yang et al. 2015). Consistent with this, we show in recently published work that B1a cells express more GFP than B2 cells and moreover that B1a cells with PtC-specific BCRs have the highest levels of Nur77-eGFP expression (Huizar et al., 2017). The latter result is shown in Author response image 9, excerpted from that paper for ease of reference. As suggested by the reviewer, this data essentially compares the repertoire of GFP-low and GFPhigh B1a cells and reveals marked enrichment of PtC-specific BCRs among GFP^hi^ B1a cells.

**Author response image 9. respfig9:** Nur77-eGFP expression identifies selfreactive PtC-specific B-1a cells in vivo. (Figure 2A,B reproduced from Figure 2 of Huizar et al., 2017, Immunohorizons, published under the Creative Commons Attribution 4.0 International Public License (CC BY4.0; https://creativecommons.org/licenses/by/4.0/)). (**A**) PerC cells from Nur77-eGFP reporter mice were stained immediately ex vivo on ice with surfacemarkers and PtC-rhodamine liposomes to identify B cell subsets. Representativeplots show PtC+ cells in B-2 and B-1a PerC subsets. (**B**) Representative histogramsdepict GFPexpression in PtC+ and PtC- B-1a cells.

7) As cited by the authors, a link between BCR signal strength and cell fate has been established some years ago, in which weak BCR signals will allow MZ B cell generation but prevent B1 cell generation. In contrast, strong signals might favor B1 generation/maintained. No major differences were observed in B1 or MZ cell generation between IgM-only, IgD-only and WT. However, under competitive conditions, IgD-only B cells fill up preferentially the MZ B cell compartment and are unable to differentiate to B1a cells. How are these cells phenotypically?

Please see response to point 6 above as well as Figure 4—figure supplement 1 demonstrating cell fate differences even in the absence of competition. See also Author response image 8 which specifically depicts MZ and B1a phenotypes.

MZ are BM-derived while B1a cells are mainly fetal liver derived and long lived. It has been proposed that in B2 cells IgD provide survival signals. Is it possible that B1 cells required survival signals via IgM and that IgD cannot provide survival signals or self-renewal signals? Is it really a matter of signal strength or of signal quality? How is this piece of data related to autoantigen recognition?

Our data does not directly distinguish between generation and survival of B1a cells. We have clarified our manuscript text to make this explicit (subsection “Cell-intrinsic skewing of B cell development by the IgM and IgD BCRs”, end of first paragraph). However, antigen-dependent BCR signal strength is essential for normal steady-state numbers of B1a cells in mature mice (despite their fetal origin) as described in literature cited in our manuscript. It is highly likely that such strong BCR signaling is antigen-dependent at least in part because the B1a repertoire is highly self-reactive. Therefore, it is reasonable to argue that impaired antigen-dependent signaling correlates with and likely accounts for loss of IgD-only B1a cells in IgM-deficient mice and in IgM^−/−^ mice. Our in vivo data does not distinguish between signal strength and “signal quality”. Our revised manuscript addresses this point explicitly (see response to point 4 above).

8) Which bibliography supports the following sentence "endogenous auto-antigens may differ significantly from model antigens"? The authors used HEL as model antigen and as endogenous auto-antigen, is the model antigen HEL not differing from endogenous auto-antigens?

We study soluble HEL model antigen as a starting point to validate function of our Nur77eGFP reporter, but our ambition in this study was really to understand how the IgM and IgD isotypes in a diverse repertoire of B cells respond to bona fide endogenous antigens.

The 9G4 anti-idiotype antibody recognizes naïve human B cells expressing the IgHV4-34 heavy chain segment. These cells account for 5-10% of circulating naïve human B cells, have been reported to recognize the I/i blood group antigen expressed on the surface of circulating red blood cells as well as O-linked carbohydrates on the B220 isoform of CD45, and exhibit downregulation of IgM consistent with self-reactivity. These bona fide endogenous antigens differ from model antigens such as HEL by their chemical structure (carbohydrate vs. protein) and form (cell surface vs. soluble), and likely by their affinity for the BCR. We have presented this example explicitly in both the Introduction (fourth paragraph) and Discussion (fourth paragraph) of our revised manuscript along with relevant citations (Quach et al., 2011; Reed et al., 2016).

9) Similar to the results obtained in vitro, the results in the Lyn^−/−^ mice support that both IgM and IgD-BCR provide sufficient signals for B cell activation (CD86, CD69) and for GC differentiation against the theory of the authors that "IgD-BCR sensing" of antigen is impaired.

IgD is not incapable of transmitting allantigen-dependent signals. Rather, we argue that it does so less efficiently than IgM especially in response to endogenous antigens. Indeed, if this were not the case, IgD would not be able to mediate B cell development and immune responses as reported in the original description of IgM^−/−^ mice by Frank Brombacher and colleagues (Lutz et al., 1998). Rather, we argue that IgD-only B cells are particularly defective at adopting cell fates requiring strong BCR signaling. Our manuscript has identified a range of in vivo phenotypes which IgD alone is insufficient to drive (e.g. high Nur77-eGFP expression, B1a cell fate, unswitched PC expansion in Lyn^−/−^ mice, autoantibodies in Lyn^−/−^ mice). By contrast, GC fate is not compromised. This is consistent with intact affinity maturation in response to TD-immunogens as reported by Brombacher and colleagues. We propose this is because weak BCR signals are sufficient to drive GC fate, in contrast to SLPC fate. Indeed, there is an extensive literature supporting this model (cited in our manuscript). Similarly, we argue that BCR signaling transmitted through IgD in the absence of Lyn is sufficient to upregulate activation markers and GC fate, but apparently not sufficient to drive unswitched PC or autoantibody production. The model we propose is that IgD-dependent signaling in vivo in response to endogenous antigens is not completely defective, but is rather less sensitive than that mediated by IgM. In other words, our paper describes which BCR signaling thresholds IgD can overcome and which ones it cannot.

10) Regarding the production of autoantibodies in Figure 5, why the production of anti-dsDNA antibodies is also increased in the WT cells that co-exist with the IgM-only B cells? Indeed, at 25 weeks the basal levels are elevated in all the mice compared to all the other situations.

We have noted a reduced penetrance of autoantibody production in IgM^−/−^ Lyn^−/−^ mice relative to IgD^−/−^ Lyn^−/−^ mice. In order to compensate for this, we collected additional serum samples from *IgM^−/−^ Lyn^−/−^* mice aged beyond 24 weeks (between initial manuscript submission and current revision included as *new* data in this revision). This allowed us to obtain more samples with detectable anti-dsDNA IgG2a and validate our finding that IgDonly cells are indeed protected from giving rise to anti-dsDNA IgG2a-secreting plasma cells relative to WT B cells in the same animal (Figure 5E). We believe these additional data further strengthen our conclusions. Our favored explanation for the reduced penetrance of autoantibodies in IgM^−/−^ Lyn^−/−^ mice relative to IgD^−/−^ Lyn^−/−^ mice is that the frequency of cells competent to initiate anti-dsDNA IgG2a PC production regulates disease progression. This was recently shown to be important for GC initiation in an autoimmune mouse model (Degn et al., 2017). We speculate that it might also play a role in PC differentiation, either through generation of an inflammatory environment or through a feed-forward mechanism where circulating autoantibodies drive further autoantibody generation. In addition, non-cell-autonomous differences in serum IgM levels between may contribute.

It is possible that other receptors, such as germline-encoded receptors, cooperate only with IgM-BCR to break tolerance and not with IgD? This will be independent of BCR strength but dependent of crosstalk between receptors. This crosstalk has been described at least for TLR4/IgM-BCR and RP105/IgM-BCR.

We understand that the reviewer is referring to the following publications:

Otipoby, K. L., et al., 2015. The B-cell antigen receptor integrates adaptive and innate immune signals.Proc Natl Acad Sci U S A 112: 12145-12150.

Ogata et al., 2000. The Toll-like Receptor Protein Rp105 Regulates Lipopolysaccharide Signaling in B Cells. J Exp Med. 2000 Jul 3; 192(1): 23–30.

Schweighoffer E, Nys J, Vanes L, Smithers N, Tybulewicz VLJ. 2017. TLR4 signals in B lymphocytes are transduced via the B cell antigen receptor and SYK. The Journal of experimental medicine214:1269-1280

TLR4 signals in response to LPS. The TLR protein RP105 also regulates responses to LPS in B cells. We show new data to establish that IgD-only as well as IgM-only B cells respond efficiently to LPS as well as other canonical TLR ligands (new Figure 1—figure supplement 3A). Therefore, these signals do not uniquely cooperate with the IgM or the IgD BCR and would not account for phenotypes described in our manuscript. We address this explicitly in our revised manuscript (Discussion, sixth paragraph).

11) In Figure 6A, how are the levels of Ets1 in IgM-only? And in Figure 6B, how is intracellular IgD in WT or IgM-only mice?

Based on how Figure 6A is presented, we assume that the reviewer meant to ask about Ets1 levels in IgD-only (*IgM^−/−^ Lyn^+/+^*) B cells. We generated additional lysates and found that IgD-only B cells have normal Ets1 levels relative to WT B cells at steady state (Author response image 10). We did not detect appreciable numbers IgD^+^ plasma cells in WT and IgM-only mice.

**Author response image 10. respfig10:** Ets1 protein expression is unaltered in splenic B cells the absence of IgD. Representative western blot of splenic B cells probed for Ets1 on left. Graph depicts mean normalized protein expression -/- SEM of N=3 biological replicates.

I do not agree with the interpretation of Figure 6E, IgM^−/−^Lyn^−/−^ have elevated IgD serum levels when compared to Lyn^−/−^.

The message of Figure 6 is that WT B cells exhibit massive expansion of unswitched plasma cells on the Lyn^−/−^ background, while IgD-only B cells do not and are protected from this phenotype. We believe that the reviewer has made a typo and misinterpreted our claims because we do not show IgD levels in *Lyn^−/−^*. Rather, in Figure 6E we compare serum IgD between IgM^−/−^ Lyn^−/−^ and IgM^−/−^. The point of this figure is to show that there is no difference in serum IgD between these genotypes; the dilution curves between the two genotypes are overlaid across the linear part of the curve showing concentration of IgD is unaltered between these genotypes. This corresponds to lack of expansion of IgD plasma cells in IgM^−/−^ Lyn^−/−^ mice (Figure 6B, C). This is in contrast to expansion of IgM plasma cells and increased serum IgM (approximately 10x) in Lyn^−/−^ mice (Figure 6B, C, D). We believe the reviewer mistakenly thought we were claiming that there was a difference.

12) Why are IgD-only less competitive to enter the GC but disfavored to produced SLPC responses? Is this because BCR-strength as proposed by the authors or by the effect of isotype-specific interactions with co-receptors?

See above response to points 4 and 9.

Reviewer #3:In this manuscript the authors present an extensive body of work comparing the development and responsiveness of polyclonal B cells capable of expressing either IgM or IgD BCRs prior to undergoing class switching. Whilst there is an extensive literature comparing the functional capabilities of these two (normally) co-expressed classes of BCR, this particular study examines a variety of readouts not previously examined, in particular the apparent sensing of endogenous self-antigens. Despite some concern about the different strain origins of the original IgM and IgD KO lines, the authors appear to control for this adequately throughout the study.Overall I found the experiments to be well-performed and thoughtfully interpreted. The data presented here go some way to resolve recent controversies in the field and as such provide a valuable source of new information to help resolve the longstanding question of why naive, resting B cells co-express IgM and IgD BCRs.I have just a couple of specific points:1) In the SRBC-induced responses, did the IgM KO mice possess a general deficiency in PC production apart from IgG1-switched PCs e.g. in producing unswitched PCs?2) Similarly, was the production of switched (e.g. IgG1+) GC B cells different between the IgM-only and IgD-only B cells?It would be important to add this information to clarify the nature of the defect involved.

We have examined IgM and IgD PC generated in response to SRBC immunization (new Figure 7—figure supplement 1C, D). We find that both IgM-only and IgD-only mice can generate unswitched PCs. IgD-only B cells and WT B cells produce similar numbers in absence of competition, while IgD-only B cells produce more in competitive setting. This trend correlates very well with the relative numbers of MZ B cells of each genotype (i.e. expanded number of IgD-only B cells especially in IgM+/- mice –Figure 4—figure supplement 1B and Figure 4D). MZ B cells efficiently generate short-lived unswitched plasma cells even in response to TD-antigens (Phan et al., 2005; Song and Cerny, 2003). We therefore suspect that IgG1 PCs emanate from the Fo B cell compartment, while the unswitched PC response may originate in the MZ B cell compartment. Normal SLPC generation by IgM-deficient MZ B cells in response to immunization would be consistent with previously reported normal T-independent II responses in IgM-deficient mice (Lutz et al., 1998). However, we cannot exclude a contribution of follicular-derived PCs that have failed to class switch to IgG1. We have modified our manuscript accordingly (subsection “IgM BCR is required for efficient generation of short-lived IgG1^+^ plasma cells but is dispensable for GC B cell fate”, third paragraph).

In pilot experiments at the time point analyzed, we observed very low numbers of switched GC B cells in all genotypes in response to sheep exposed on the membranes of dying cells RBC immunization; the vast majority were unswitched (see the second paragraph of the aforementioned subsection).

[Editors' note: the author responses to the re-review follow.]

This manuscript provides novel insights into different functions between IgM and IgD BCRs. There are some remaining issues that need to be addressed before acceptance, as outlined below in the separate reviews. Specifically, we request that you recognize that the Nur77-GFP signals in B cells might be the result of BCR stimulation by a broader collection of non-inflammatory antigens in vivo, not only self-antigens. They might be globally named as endogenous antigens but not exclusively self-antigens, or endogenous self-antigens.

We are extremely grateful for the thoughtful editorial and review process. We have addressed all reviewer queries. Specifically, we have incorporated the possibility that Nur77-eGFP expression downstream of BCR engagement may be triggered not only by “self” antigens, but also by non-inflammatory antigens that are not germline encoded. We have made this clear in our revised manuscript. Please see response to reviewer #3 below for detailed description of manuscript edits that address this point.

Reviewer #3:I have read with enthusiasm and interest the revised version of the manuscript entitled "Differential sensing of endogenous antigens and control of B cell fate by IgM and IgD B cell receptors" by Noviski et al. I appreciate the determination of the investigators to address the issues that arose during the revision process and thank the authors for their efforts.One of my major points in the previous revision round was whether the Nur77-GFP positive B cells were indeed positive because activation of the BCR by self-antigens (as claimed by the authors) or by a combination of signals by the BCR and/or additional receptors expressed on the surface of B cells.The authors have appropriately answered part of this question by showing that some of these receptors do not activate expression of Nur77 (TLR3, TLR7, TLR9, CXCR4). Other receptors, such as TLR4, TLR9 and TLR1/2 do activate expression of Nur77-GFP (Author response image 6). However, the authors demonstrate here (Figure 1—figure supplement 3) that upregulation of Nur77-GFP by these receptors is equal between IgM-only and IgD-only B cells (although the concentration of ligands chosen are too high).

We have performed these assays with lower doses of LPS and CpG and see comparable upregulation of Nur77-eGFP as well as activation markers by IgM-/- and IgD-/- B cells. Lower doses of Pam3CSK4 were insufficient to activate B cells or upregulate Nur77-eGFP. We also corrected an error: CpG dose units are nM, not µM. We have now revised Figure 1—figure supplement 3 to depict lower doses of stimuli.

The authors also show in the new version, that endogenous levels of Nur77 transcripts in B cells are equally elevated between Germ-free mice and SPF, arguing that Nur77 signals in B cells do not from microbial stimulation (although these signals might come from the endotoxin present in the chow, or did the authors used endotoxin-free chow?). Moreover, the levels of endogenous Nur77 transcripts were equally elevated between MyD88KO and MyD88WT peritoneal B1a cells. Here, I would like to ask why peritoneal B1a cells were chosen and not splenic B cells as in Figure 1—figure supplement 1D-E?

We have now modified Figure 1—figure supplement 1 to include qPCR data analyzing *Nr4a1* transcript in whole splenocytes taken ex vivo from B cell-conditional MyD88KO and MyD88WT mice (Figure 1—figure supplement 1G).

Since submitting our revised manuscript to *eLife* we optimized our *Nr4a1* primer design to improve specificity. This lowers non-specific background signal in *Nr4a1-/-* samples and has enabled us to detect endogenous *Nr4a1* transcript directly ex vivo with greater sensitivity and dynamic range. Therefore, we reanalyzed *Nr4a1* transcript expression in germ-free and SPF as well as B cell-conditional MyD88 KO and WT splenocytes alongside *Nr4a1-/-* controls. We have replaced Figure 1—figure supplement 1E to display *Nr4a1* transcript in germ-free and SPF splenocytes analyzed immediately ex vivo, and we replaced Figure 1—figure supplement 1D to depict endogenous protein in splenic B cells from the same mice also analyzed directly ex vivo.

Figure 1—figure supplement 1D-G now collectively show no statistically significant differences in Nur77 protein or *Nr4a1* transcript analyzed directly ex vivo between germ-free and SPF mice, as well as between B cell-conditional MyD88 KO and WT samples.

However, I do not agree that the Nur77-GFP signals in B cells are the consequence of self-antigen (self-antigen defined by an antigen encoded by the DNA of the self-organism) stimulation by the BCR. I disagree with the authors using self-antigen (subsection “Cell-intrinsic skewing of B cell development by the IgM and IgD BCRs”, first paragraph), endogenous auto-antigen (subsection “Either IgD or IgM BCR is sufficient to mediate polyclonal B cell activation and germinal center differentiation in Lyn^−/−^ mice”, first paragraph) and antigen encounters in vivo (subsection “IgD BCRs sense endogenous antigens less efficiently than IgM BCRs”) as synonymous. I fully agree that Nur77-GFP signals are upregulated upon BCR stimulation by antigen and that antigen is necessary to drive Nur77-GFP expression in vivo. But I do not agree that self-antigen is necessary to drive Nur77-GFP expression in vivo. My arguments are:• Nur77-GFP signals are highly upregulated when B cells leave the sterile/defined microenvironment of the BM. Nur77-GFP expression timely occurs when cells encounter "antigens" in the periphery, and those antigens activate the BCR. But these antigens are not necessarily self-antigens, they can be food-antigens, LPS (from the chow, antibodies against LPS are broadly recognized), and other non-pathological or non-inflammatory antigens recognized by the BCR. These antigens might activate B cells to upregulate Nur77, but in the absence of T cell help or an inflammatory milieu will not drive an immune response.• The results with the IgHEL BCR Tg mice (Author response image 1) are fully compatible with the above-described scenario. These mice express a clonal BCR not able to recognize any self-antigen, but also, not able to recognize any of the possible non-inflammatory antigens that will be encountered in the periphery, for this reason the signal of Nur77 is lower. This interpretation is also compatible with higher level of Nur77-GFP in HEL Tg cells exposed to cognate HEL antigen that in WT. Related to this, the results show in Author response image 6 are also compatible with this scenario since Nur77-GFP upregulation is reduced in IgHEL BCR Tg mice in respond to LPS (small reduction), CpG (clear at lower concentration), and to Pam3CSK4 (visible at both concentrations). A possible interpretation of these results is that the HEL Tg BCR is not able to recognize epitopes in those substances that, together with the corresponding TLR, aid to activate Nur77-GFP in B cells.• The results with the VH3H9 Tg mice (Author response image 3) are also fully compatible with the above-described scenario, since the κ+ cells (not recognizing endogenous DNA) also up-regulate (even to the same levels than λ+ cells) Nur-77 upon leaving the BM. Most probably because they encounter non-inflammatory antigens in the periphery.Taken together, I fully agree that the present manuscript is improved upon revision and most of my requests have been clarify. However, I still request that the authors recognize that the Nur77-GFP signals in B cells might be the result of BCR stimulation by a broader collection of non-inflammatory antigens in vivo, not only self-antigens. They might be globally named as endogenous antigens but not exclusively self-antigens, or endogenous self-antigens. The main conclusion of the manuscript is unaltered, namely that IgD-BCR signals differently upon encounter of non-inflammatory antigens in vivo, but to narrow this conclusion only to self-antigens is not proven and not justified.

We gratefully acknowledge the possibility that endogenous antigens that signal through the BCR of naïve B cells in vivo may not exclusively represent “self” antigens encoded by the host genome or generated by enzymes encoded by the host genome. We have therefore modified the text of our revised manuscript to carefully eliminate such an implication. We have now consistently used the term “endogenous” rather than “self” antigens in order to encompass non-inflammatory exogenous antigens (such as food antigens) that may signal through the BCR. Specifically, we have modified citations in the first paragraph of the subsection “Cell-intrinsic skewing of B cell development by the IgM and IgD BCRs” and “Either IgD or IgM BCR is sufficient to mediate polyclonal B cell activation and germinal center differentiation in *Lyn-/-* mice”. We have gone on to also edit references throughout the manuscript in order to clarify the same point. Finally, we have made this point explicitly in our revised Discussion:

“These antigens may not be exclusively restricted to those that are germline-encoded or generated by host enzymes; rather they may include non-inflammatory antigens taken up through the gastrointestinal tract and recognized by specific BCRs.”

References:

Baumgarth, N. 2016. B-1 Cell Heterogeneity and the Regulation of Natural and Antigen-Induced IgM Production. Frontiers in immunology 7:324.

Bendelac, A., M. Bonneville, and J.F. Kearney. 2001. Autoreactivity by design: innate B and T lymphocytes. Nat Rev Immunol 1:177-186.

Cancro, M.P., and J.F. Kearney. 2004. B cell positive selection: road map to the primary repertoire? J Immunol 173:15-19.

Cariappa, A., and S. Pillai. 2002. Antigen-dependent B-cell development. Curr Opin Immunol 14:241-249.

Degn, S.E., C.E. van der Poel, D.J. Firl, B. Ayoglu, F.A. Al Qureshah, G. Bajic, L. Mesin, C.A. Reynaud, J.C. Weill, P.J. Utz, G.D. Victora, and M.C. Carroll. 2017. Clonal Evolution of Autoreactive Germinal Centers. Cell 170:913-926 e919.

Erikson, J., M.Z. Radic, S.A. Camper, R.R. Hardy, C. Carmack, and M. Weigert. 1991. Expression of anti-DNA immunoglobulin transgenes in non-autoimmune mice. Nature 349:331-334.

Fields, M.L., B.D. Hondowicz, G.N. Wharton, B.S. Adair, M.H. Metzgar, S.T. Alexander, A.J. Caton, and J. Erikson. 2005. The regulation and activation of lupus-associated B cells. Immunological reviews 204:165-183.

Freitas, A.A., M.P. Lembezat, and A. Coutinho. 1989. Expression of antibody V-regions is genetically and developmentally controlled and modulated by the B lymphocyte environment. Int Immunol 1:342-354.

Gu, H., D. Tarlinton, W. Muller, K. Rajewsky, and I. Forster. 1991. Most peripheral B cells in mice are ligand selected. J Exp Med 173:1357-1371.

Hayakawa, K., M. Asano, S.A. Shinton, M. Gui, L.-J. Wen, J. Dashoff, and R.R. Hardy. 2003. Positive selection of anti-thy-1 autoreactive B-1 cells and natural serum autoantibody production independent from bone marrow B cell development. J Exp Med 197:87-99.

Huizar, J., C. Tan, M. Noviski, J.L. Mueller, and J. Zikherman. 2017. Nur77 Is Upregulated in B-1a Cells by Chronic Self-Antigen Stimulation and Limits Generation of Natural IgM Plasma Cells. Immunohorizons 1:188-197.

Kraus, M., M.B. Alimzhanov, N. Rajewsky, and K. Rajewsky. 2004. Survival of Resting Mature B Lymphocytes Depends on BCR Signaling via the Igα/β Heterodimer. Cell 117:787-800.

Lam, K.P., R. Kuhn, and K. Rajewsky. 1997. In vivo ablation of surface immunoglobulin on mature B cells by inducible gene targeting results in rapid cell death. Cell 90:1073-1083.

Levine, M.H., A.M. Haberman, D.B. Sant'Angelo, L.G. Hannum, M.P. Cancro, C.A. Janeway, Jr., and M.J. Shlomchik. 2000. A B-cell receptor-specific selection step governs immature to mature B cell differentiation. Proc Natl Acad Sci U S A 97:2743-2748.

Lutz, C., B. Ledermann, M.H. Kosco-Vilbois, A.F. Ochsenbein, R.M. Zinkernagel, G. Kohler, and F. Brombacher. 1998. IgD can largely substitute for loss of IgM function in B cells. Nature 393:797-801.

Moran, A.E., K.L. Holzapfel, Y. Xing, N.R. Cunningham, J.S. Maltzman, J. Punt, and K.A. Hogquist. 2011. T cell receptor signal strength in Treg and iNKT cell development demonstrated by a novel fluorescent reporter mouse. The Journal of experimental medicine 208:1279- 1289.

Phan, T.G., S. Gardam, A. Basten, and R. Brink. 2005. Altered migration, recruitment, and somatic hypermutation in the early response of marginal zone B cells to T celldependent antigen. J Immunol 174:4567-4578.

Quach, T.D., N. Manjarrez-Orduno, D.G. Adlowitz, L. Silver, H. Yang, C. Wei, E.C. Milner, and I. Sanz. 2011. Anergic responses characterize a large fraction of human autoreactive naive B cells expressing low levels of surface IgM. J Immunol 186:4640-4648.

Reed, J.H., J. Jackson, D. Christ, and C.C. Goodnow. 2016. Clonal redemption of autoantibodies by somatic hypermutation away from self-reactivity during human immunization. The Journal of experimental medicine 213:1255-1265.

Rosado, M.M., and A.A. Freitas. 1998. The role of the B cell receptor V region in peripheral B cell survival. Eur J Immunol 28:2685-2693.

Sabouri, Z., S. Perotti, E. Spierings, P. Humburg, M. Yabas, H. Bergmann, K. Horikawa, C. Roots, S. Lambe, C. Young, T.D. Andrews, M. Field, A. Enders, J.H. Reed, and C.C. Goodnow. 2016. IgD attenuates the IgM-induced anergy response in transitional and mature B cells. Nat Commun 7:13381.

Sabouri, Z., P. Schofield, K. Horikawa, E. Spierings, D. Kipling, K.L. Randall, D. Langley, B. Roome, R. Vazquez-Lombardi, R. Rouet, J. Hermes, T.D. Chan, R. Brink, D.K. Dunn-Walters, D. Christ, and C.C. Goodnow. 2014. Redemption of autoantibodies on anergic B cells by variable-region glycosylation and mutation away from self-reactivity. Proc Natl Acad Sci U S A 111:E2567-2575.

Song, H., and J. Cerny. 2003. Functional heterogeneity of marginal zone B cells revealed by their ability to generate both early antibody-forming cells and germinal centers with hypermutation and memory in response to a T-dependent antigen. The Journal of experimental medicine 198:1923-1935.

Su, T.T., B. Guo, B. Wei, J. Braun, and D.J. Rawlings. 2004. Signaling in transitional type 2 B cells is critical for peripheral B-cell development. Immunol Rev 197:161-178.

Thomas, M.D., B. Srivastava, and D. Allman. 2006. Regulation of peripheral B cell maturation. Cell Immunol 239:92-102. Wang, H., and S.H. Clarke. 2003. Evidence for a ligand-mediated positive selection signal in differentiation to a mature B cell. J Immunol 171:6381-6388.

Wardemann, H., S. Yurasov, A. Schaefer, J.W. Young, E. Meffre, and M.C. Nussenzweig. 2003. Predominant autoantibody production by early human B cell precursors. Science 301:1374-1377.

Yang, Y., C. Wang, Q. Yang, A. Kantor, H. Chu, E. Ghosn, G. Qin, S. Mazmanian, J. Han, and L. Herzenberg. 2015. Distinct mechanisms define murine B cell lineage immunoglobulin heavy chain (IgH) repertoires. eLife 4:e09083.

Zikherman, J., R. Parameswaran, and A. Weiss. 2012. Endogenous antigen tunes the responsiveness of naive B cells but not T cells. Nature 489:160-164.